# The influence of temporal context on vision over multiple time scales

Kacie Lee[1], Reuben Rideaux[1,2]*

[1]School of Psychology, The University of Sydney, Sydney, Australia; [2]Queensland Brain Institute, The University of Queensland, Brisbane, Australia

## eLife Assessment

This **fundamental** study shows how past experiences shape perception across short, medium, and long time scales, using a single behavioural paradigm and reanalysed EEG data. It provides **convincing** evidence for two processes across all scales: an attention-dependent mechanism that speeds responses to expected events, and an attention-independent mechanism where expected events are encoded less precisely, consistent with feedforward dampening. The work offers a unifying account of temporal context effects, though stronger brain-behaviour links, integration with serial dependence attraction and repulsion models, and extension to other timescale definitions would further strengthen the contribution.

*For correspondence:
reuben.rideaux@sydney.edu.au

Competing interest: The authors declare that no competing interests exist.

## Abstract

Past sensory experiences influence perception of the present. Multiple research subfields have emerged to study this phenomenon at different temporal scales. These phenomena fall into three categories: the influence of immediately preceding sensory events (micro), expectations established by short sequences of events (meso), and regularities over long sequences of events (macro). In a single paradigm, we examined the influence of temporal context on human perception at each scale. By integrating behavioral and pupillometry recordings with electroencephalographical recordings from a previous study, we identify two distinct mechanisms that operate across all scales. The first is moderated by attention and supports rapid motor responses to expected events. The second operates independently of task demands and dampens the feedforward neural responses produced by expected events, leading to unexpected events eliciting earlier and more precise neural representations.

## Introduction

The capacity to adapt to patterns in the environment supports biological function from sensory processing to motor action. The temporal context in which sensory events are embedded can be leveraged to more effectively process and respond to this information. For example, expert tennis players predict the trajectory of the ball from the movements of their opponent and use this information to prepare their return (*Wolpert and Flanagan, 2001*). Influential theories of normative brain function such as predictive coding posit that temporal context serves to improve representational fidelity and reduce neurometabolic expenditure (*Barlow, 1961*; *Friston, 2005*; *Rao and Ballard, 1999*).

Humans are sensitive to the temporal context of events across multiple time scales. At the shortest scale, each event influences processing of the next event (*Figure 1a*, micro). For example, stimulus reproductions are biased towards previous stimuli. This phenomenon is referred to as *serial dependency* and has been demonstrated for a range of visual and auditory features (*Cicchini et al., 2014*; *Cicchini et al., 2024*; *Fischer and Whitney, 2014*). At intermediate scales, short sequences of events form patterns that uniquely influence responses to subsequent events (*Figure 1a*, meso). For instance,

**Figure 1.** Multiple scales of temporal context. (**a**) Illustration of three scales of temporal context in a binary time series. At the micro scale, events can either stay the same over time (repeat; R) or change (alternate; A). At the meso scale, short sequences of events (e.g. 5) form patterns that vary in their regularity (e.g. four repeats of the same event seems more regular than a mixture of repeats and alternations between events). At the macro scale, general trends regarding the relative frequency of different events are formed over longer time periods. (**b**) To test the influence of temporal context on visual perception across different scales, participants were instructed to indicate the location of serially presented targets, which were randomly positioned on an imaginary circle centered on fixation. Participants performed a speeded binary judgement (e.g. left/right of fixation) on each trial and additionally reproduced the location of the target on 10% of trials. Trials were categorized as either (**c**, top) repeat (R) or (**c**, bottom) alternate (A) based on the location of the target relative to the previous target, according to three spatial reference planes: task-related (light cyan), task-unrelated (dark cyan), and stimulus-related (orange). (**d**) In Experiment 1, the location of targets was uniformly sampled such that repeat and alternate trials were equally likely. In Experiment 2, the probability of repeat and alternate trials was biased across the task-related and unrelated planes.

events that satisfy regular patterns produce attenuated neural responses (*oddball effect*; *Squires et al., 1975*) and are responded to more quickly and accurately than events that violate them (*sequential dependencies*; *Kirby, 1976*; *Remington, 1969*). At longer time scales, the relative frequency of past events can alter how those in the present are processed (*Figure 1a*, macro). These effects are often referred to as *statistical learning* and are thought to reflect adaptation to regularities in the environment (*Schapiro and Turk-Browne, 2015*; *Simoncelli and Olshausen, 2001*).

Incoming sensory information is shaped by its temporal context across time scales that range many orders of magnitude. Previous work has developed a variety of experimental designs to isolate the influence of temporal context at each scale and examine its behavioral and neural consequences. This approach has propagated multiple fields of research and implicitly asserts qualitative differences; however, it limits comparison between temporal context at different scales. It is possible that temporal context serves perception differently at each time scale and is thus associated with unique adaptive influences on sensory processing. For example, perception may be best supported by one mechanism that promotes a set of outcomes from moment to moment and another mechanism that promotes a different set of outcomes in response to trends formed over long periods. Alternatively, a common rule may be applied at all scales of temporal context, signaling a unifying adaptive mechanism. Predictive coding theory offers a normative framework with which to understand the influence of temporal context across scales, but contradictory evidence (*Friston, 2005*; *Kok et al., 2012*; *de Lange et al., 2018*; *Richter et al., 2022*; i.e. sharpening or dampening of expected events) from distinct experimental paradigms has challenged this unification.

We designed a novel paradigm to investigate how temporal context shapes perception across multiple scales. We further used electroencephalography (EEG) and pupillometry recordings to characterize the neural mechanisms associated with these perceptual consequences. We identify two

distinct mechanisms that operate across all scales. The first is moderated by attention and supports rapid motor responses to expected events. The second is independent of task demands and dampens the feedforward neural responses to expected events, leading to unexpected events eliciting earlier and more precise neural representations. Together, these adaptive mechanisms explain a raft of temporal sensory phenomena and provide evidence for a dampening account of predictive coding.

## Results

To measure changes in visual perception associated with different scales of temporal context, we tasked participants with indicating the location of serially presented visual stimuli (Gaussian blobs randomly positioned at a fixed distance around a central fixation point). To assess response time and task accuracy, participants performed a speeded binary judgement (e.g. left or right of fixation) on each trial. On 10% of trials, participants additionally reproduced the location of the stimulus, providing a measure of recall precision. To test the influence of attention, trials were sorted according to two spatial reference planes, based on the location of the stimulus: task-related and task-unrelated (*Figure 1b*). The task-related plane corresponded to participants' binary judgement (*Figure 1b*, light cyan vertical dashed line) and the task-unrelated plane was orthogonal to this (*Figure 1b*, dark cyan horizontal dashed line). For example, if a participant was tasked with performing a left-or-right-of-fixation judgement, then their task-related plane was the vertical boundary between the left and right side of fixation, while their task-unrelated plane was the horizontal boundary. The former (left-right) axis is relevant to their task while the latter (top-bottom) axis is orthogonal and task irrelevant. This

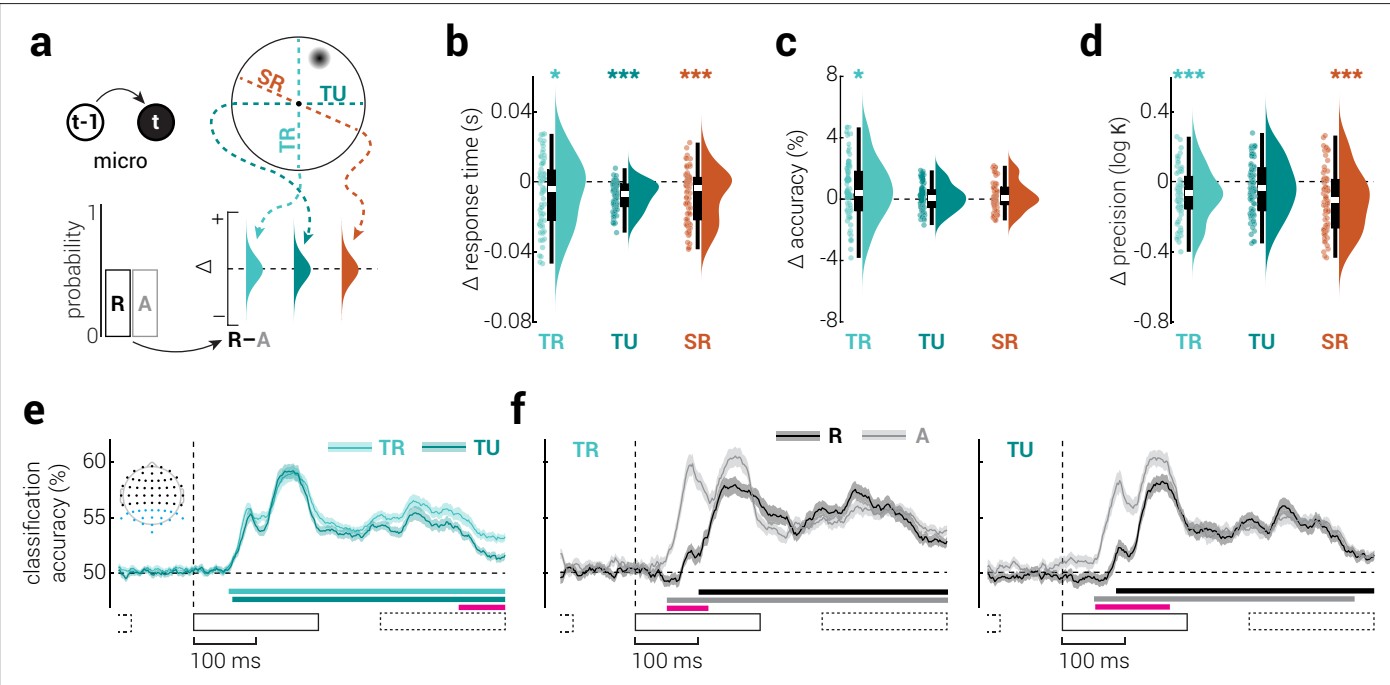

**Figure 2.** The influence of micro-scale temporal context on visual processing. (**a**) Micro temporal context refers to the influence of the last event on the current event. We assessed its influence by comparing task performance for repeat and alternate presentations along task-related (light cyan), task-unrelated (dark cyan), and stimulus-related (orange) reference planes. (**b–d**) The difference (repeat – alternate) in (**b**) response time, (**c**) task accuracy, and (**d**) precision for the three reference planes. Asterisks indicate significant differences (*p<0.05, **p<0.01, ***p<0.001). (**e**) Classification accuracy of stimuli presented on different sides of task-related and unrelated planes, from re-analysis of previously published EEG data (*Rideaux, 2024*). (**f**) Same as (**e**), but split into repeat and alternate stimuli, along (left) task-related and (right) unrelated planes. Inset in (**e**) shows the EEG sensors included in the analysis (blue dots). Black rectangles indicate the timing of stimulus presentations (solid: target stimulus, dashed: previous and subsequent stimuli). Shaded regions indicate ± SEM. Horizontal bars indicate cluster-corrected periods of significance (cyan and greyscale: above chance classification accuracy, pink: difference).

The online version of this article includes the following figure supplement(s) for figure 2:

**Figure supplement 1.** Decoding stimulus location and Δ location from EEG recordings.

**Figure supplement 2.** Removal of general micro temporal dependencies in EEG responses.

orthogonality can be leveraged to analyze the same data twice (once according to the task-related plane and again according to the task-unrelated plane) in order to compare performance when the relative location of an event is either task relevant or irrelevant.

## Micro-scale temporal context

At the shortest temporal scale, the speed with which events are responded to can be influenced by those that immediately precede them (*Bertelson, 1961*). We investigated the influence of the previous stimulus on the current stimulus by comparing responses between trials in which the previous stimulus was presented either on the same side (repeat) or the other side (alternate) of the reference plane (*Figures 1c and 2a*). Consistent with previous work (*Bertelson, 1961*), we found faster and more accurate responses to repeat stimuli along the task-related plane (speed: $z_{75}$=2.33, p=0.020, $r$=0.27; task accuracy: $z_{72}$=2.41, p=0.016, $r$=.28; *Figure 2b–c*, light cyan). However, in contrast, we found that repeat stimuli were recalled less precisely than alternate stimuli ($z_{72}$=3.74, p=2.00e$^{-4}$, $r$=0.44; *Figure 2d*, light cyan). For the task-unrelated plane, we found faster responses for repeat stimuli, but no difference in task accuracy, and marginally higher precision for alternate stimuli (speed: $z_{75}$=6.66, p=2.76e$^{-11}$, $r$=0.77; accuracy: $z_{72}$=0.92, p=0.357, $r$=.11; precision: $z_{76}$=1.62, p=0.105, $r$=0.19; *Figure 2b–d*, dark cyan).

While the difference in response time for repeat and alternate stimuli was of similar magnitude between task-related and unrelated planes ($z_{71}$=1.15, p=0.249), the *variance* across participants was considerably higher in the former condition ($F_{1,140}$=46.26, p=2.76e$^{-10}$). The increased variability of response time differences across the task-related plane likely reflects individual differences in attention and prioritization of responding either quickly or accurately. On each trial, the correct response (e.g. left or right) was equally probable. So, to perform the task accurately, participants were motivated to respond without bias, that is without being influenced by the previous stimulus. We would expect this to reduce the difference in response time for repeat and alternate stimuli across the task-related plane, but not the task-unrelated plane. However, task-relatedness may amplify the bias towards making faster responses for repeat stimuli, by increasing attention to the identity of stimuli as either repeats or alternations (*Rideaux, 2024*). These two opposing forces vary with task engagement and strategy and thus would be expected to produce increased variability across the task-related plane. Unlike response time, we would not expect task accuracy to be modulated by task-unrelated micro temporal context, as the tasks were orthogonal. That is, while a repeat along the task-unrelated plane may increase the speed with which an (in/correct) response is made along the task-related plane, it would not be expected to change the type of response (e.g. left or right). Precision differences were similar between task-related and unrelated planes, both in terms of magnitude ($z_{70}$=1.77, p=0.076) and variability ($F_{1,138}$=0.66, p=0.417), suggesting that modulation of representational fidelity may be operationalized by a distinct mechanism to that which modifies response speed and is not moderated by attention.

These results could be interpreted as a general expectation that sensory events, for example the image of an object on the retina, are more likely to stay the same than to change (*Fischer and Whitney, 2014*). Through this lens, these findings would indicate that micro temporal context leads to faster and more accurate responses to expected events, but more precise encoding of unexpected events. As we will show, these two outcomes resonate across all levels of temporal context and point to two distinct mechanisms that unify the influence of past experiences on perception.

To test for neural correlates of micro temporal context, we re-analyzed a previously published EEG dataset in which human participants viewed visual (arc) stimuli presented at random angles around fixation, while monitoring for targets (stimuli presented at a specific location; *Rideaux, 2024*). We used multivariate linear-discriminant analysis to classify the location of stimuli according to task-related (<or > |90°| from target angle) and unrelated planes (orthogonal to task-related plane), from parietal and occipital sensors (*Figure 2—figure supplement 1*). We found that classification accuracy was higher across the task-related plane (*Figure 2e*).

In separately assessing classification accuracy for repeat and alternate stimuli, we found temporal dependencies that were due to low-frequency signals, unrelated to stimulus properties, as evidenced by their presence in results produced from data with shuffled labels (see *Figure 2—figure supplement 2* for a full description of the temporal dependencies). We removed these general temporal dependencies by subtracting the bias in the shuffled data from the original classification accuracy. For

both task-related and unrelated planes, we found that classification accuracy was higher for alternate stimuli from ~60–160ms (i.e. the first moments of reliable stimulus decoding) and that alternate stimuli could be reliably classified ~50ms earlier than repeat stimuli (*Figure 2f*). The difference in classification accuracy between repeat and alternate stimuli was the same for task-related and unrelated conditions. This neural phenomenon may explain the increased behavioral recall precision observed for alternate stimuli. The timing and task independence of this phenomenon points to a passive, massively parallel, mechanism that operates on feedforward sensory signals by suppressing responses to repeat stimuli.

Another phenomenon associated with micro temporal context is *serial dependence*, that is reproductions of events from memory are biased towards the events that immediately preceded them (*Cicchini et al., 2014*; *Fischer and Whitney, 2014*). To assess serial dependence in the behavioral data, we began by comparing behavioral performance across a third, stimulus-related, plane (*Figure 2a*, orange); whereas task planes were constant, the stimulus-related plane was defined by the location of the stimulus on the previous trial, and thus varied from trial to trial. That is, on each trial, the target is considered a repeat if it changes location by <|90°| relative to its location on the previous trial, and an alternate if it changes by >|90°|. We found a similar pattern of results as those for task-related and unrelated planes: responses were faster for stimuli that repeated ($z_{77}$=3.73, p=2.00e$^{-4}$, r=0.43; *Figure 2b*, orange) but more precise for those that alternated ($z_{76}$=4.51, p=6.34e$^{-6}$, r=0.52; *Figure 2d*, orange). The latter result is inconsistent with previous work on serial dependence, which has reported more precise judgements when subsequently presented stimuli are similar (*Cicchini et al., 2018*).

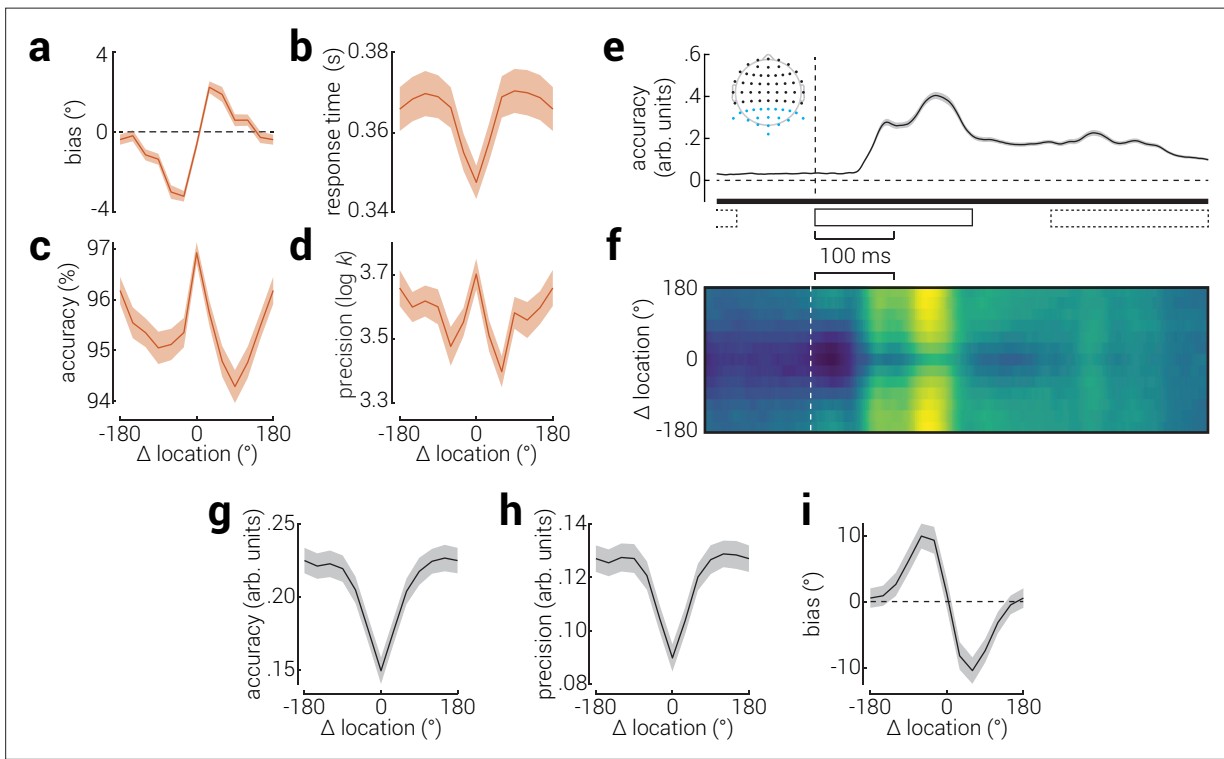

**Figure 3.** The influence of serial dependence on visual processing. Serial dependence is associated with sensory stimuli being reported as more similar to previous stimuli and is typically assessed by measuring the perception of stimuli as a function of their distance to previous stimuli (Δ location). (a) Location reproduction bias, (b) binary task response time and (c) accuracy, and (d) reproduction precision as a function of distance from the previous stimulus. (e) Decoding accuracy for stimulus location, from re-analysis of previously published EEG data (*Rideaux, 2024*). The inset shows the EEG sensors included in the analysis (blue dots), and black rectangles indicate the timing of stimulus presentations (solid: target stimulus, dashed: previous and subsequent stimuli). (f) Decoding accuracy for location, as a function of time and Δ location. Bright colors indicate higher decoding accuracy; absolute accuracy values can be inferred from (e). (g–i) Average location decoding (g) accuracy, (h) precision, and (h) bias from 50 to 500ms following stimulus onset. Horizontal bar in (e) indicates cluster-corrected periods of significance; note, all time points were significantly above chance due to temporal smear introduced by high-pass filtering (see *Figure 3—figure supplement 1* for full details). Note, the temporal abscissa is aligned across (e & f). Shaded regions indicate ± SEM.

The online version of this article includes the following figure supplement(s) for figure 3:

**Figure supplement 1.** Removal of general temporal dependencies in EEG responses for inverted encoding analyses.

To further investigate the influence of micro temporal context, we computed task performance as a function of the angular distance from the previous stimulus (Δ *location*). We calculated the bias of reproduction responses as a function of Δ location and found the archetypal pattern of attractive biases associated with serial dependence (*Figure 3a*). Consistent with our previous binary analysis, and with previous work (*Stewart, 2007*), we also found that responses were faster and more accurate when Δ location was small (*Figure 3b and c*). We further found that the precision of responses peaked at both large and small Δ locations, with the worst precision for stimuli with around ±60° Δ location (*Figure 3d*). The peak in precision for large Δ locations is consistent with alternate events being encoded more precisely, while the peak for small offsets may be explained by the attractive bias towards the previous target.

To further investigate the influence of serial dependence, we applied inverted encoding modeling to the EEG recordings to decode the angular location of stimuli. We found that decoding accuracy sharply increased from ~60ms following stimulus onset (*Figure 3e*). Note, to remove the influence of general temporal dependencies, we applied a 0.7 Hz high-pass filter to the data, which temporally smeared the stimulus-related information, resulting in above-chance decoding accuracy prior to stimulus presentation (for full details, see *Figure 3—figure supplement 1*). To understand how serial dependence influences the representation of these features, we inspected decoding accuracy as a function of both time and Δ location (*Figure 3f*). We found that decoding accuracy varied depending not only as a function of time, but also as a function of Δ location. To characterize the latter relationship, we calculated the average decoding accuracy from 50ms until the end of the epoch (500ms), as a function of Δ location (*Figure 3g*). This revealed higher accuracy for targets with larger Δ location. We found a similar pattern of results for decoding precision (*Figure 3h*). These results are consistent with the micro temporal context (behavioral) results, showing that targets that alternated were recalled more precisely. Lastly, we calculated the decoding bias as a function of Δ location and found a clear repulsive bias away from the previous stimulus (*Figure 3i*). While this result is inconsistent with the attractive behavioral bias, it is consistent with recent studies of serial dependence suggesting an initial pattern of repulsion followed by an attractive bias during the response period (*Luo et al., 2025*; *Sheehan and Serences, 2022*; *Fischer et al., 2024*).

## Meso-scale temporal context

The influence of recent events can accrue over time, producing unique changes in behavioral responses associated with short sequences of stimuli (*Figure 4a*). Higher-order sequential dependences are an example of how stimuli (at least) as far back as five events in the past can shape the speed and task accuracy of responses to the current stimulus (*Kirby, 1976*; *Remington, 1969*); however, note that these effects have been observed for more than five events (*Fritsche et al., 2020*). For five binary events (e.g. left or right), there are 16 possible sequences of repeat and alternate stimuli. Typically, sequential dependencies are assessed by computing the average response time and/or task accuracy for stimuli, binned according to which of these 16 sequences reflects their recent history. The differences observed between sequences have been interpreted as indicative of the expectation of the current stimulus, based on previous events (*Kirby, 1976*).

We tested the influence of higher-order sequential dependencies on response time, task accuracy, and recall precision, across task-related and unrelated reference planes. However, we first isolated the influence of meso temporal context from that of micro context (i.e. difference in performance between repeat and alternate stimuli), by binning responses into eight sequences formed by combining the symmetric partners in the 16-sequence array (e.g. RRRR with AAAA, RRRA with AAAR, etc.). Thus, the frequency of repeat and alternate stimuli was equal across the eight sequences and any differences observed in performance are unlikely to be due to differential responses to these events. We found that responses were faster for sequences with expected targets across both reference planes (task-related: $\chi^2_{7,469}$=375.86, p=3.56e$^{-77}$, $W$=0.84; task-unrelated: $\chi^2_{7,483}$=110.94, p=5.85e$^{-21}$, $W$=0.26; *Figure 4b*). Responses were also more accurate for expected targets across the task-related plane ($\chi^2_{7,399}$=208.37, p=2.03e$^{-41}$, $W$=0.44; *Figure 4c*). By contrast, we found no such clear relationship between precision and expectancy (both p>0.05; *Figure 4d*).

We found that micro temporal context influenced recall precision, whereas we did not detect significant modulation of precision for meso context. This may be because meso temporal effects reflect motor priming and do not precipitate visual predictions. When predictions are violated,

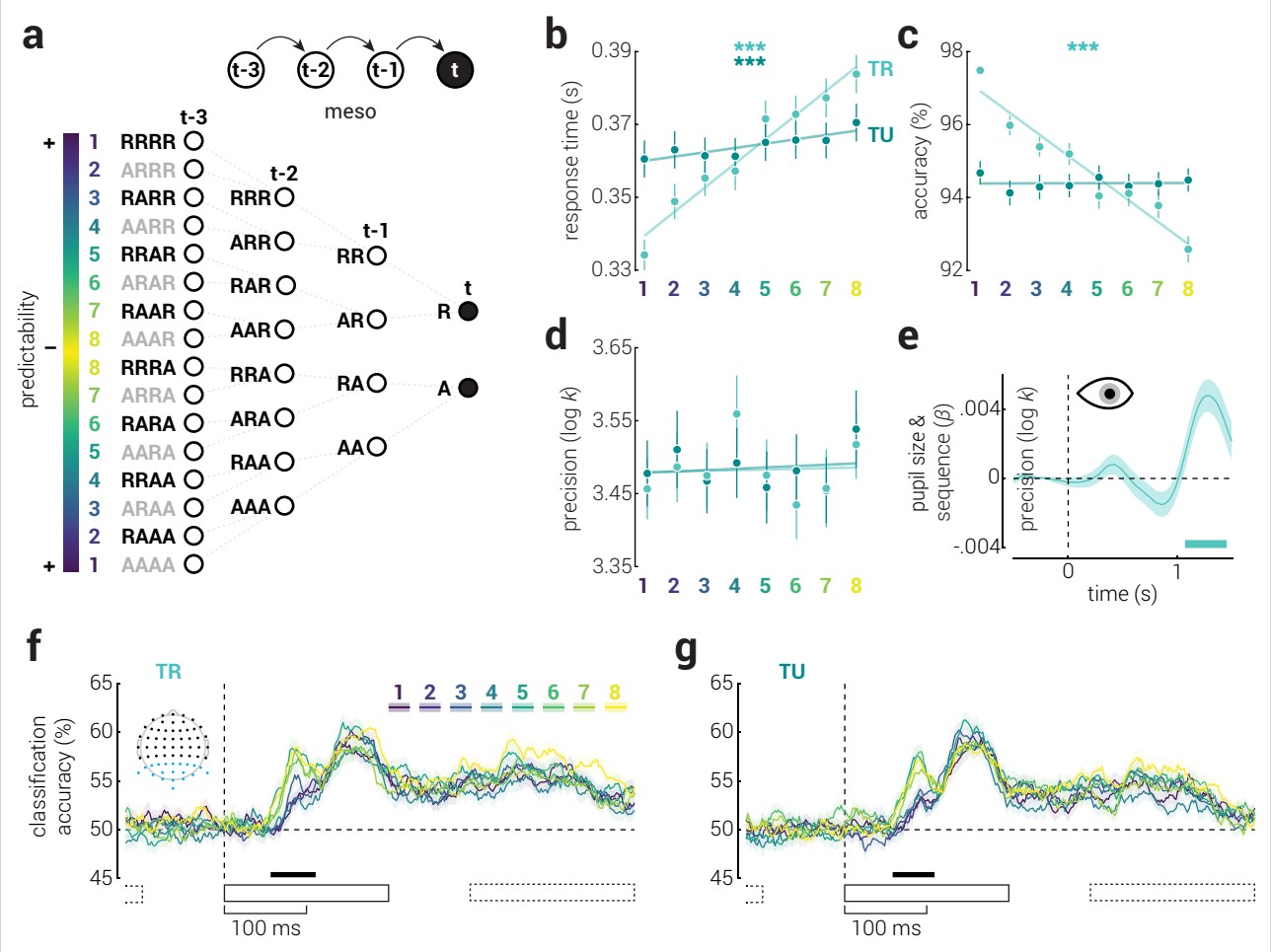

**Figure 4.** The influence of meso-scale temporal context on visual processing. (**a**) Illustration of all 16 possible sequences of repeat and alternate events for a series of five binary events. Sequences are arranged symmetrically such that those for which the final event is thought to be most expected are at the top and bottom and those for which the final event is least expected are in the middle. To control for differences between repeat and alternate events, we combined symmetric pairs, resulting in a total of eight sequences. (**b**) Response time, (**c**) task accuracy, and (**d**) precision as a function of sequence, across task-related and unrelated reference planes. Note, lower numbers on the abscissa are associated with sequences in which the final target stimulus is more expected. Asterisks indicate significant main effects of sequence (***p<0.001). Error bars indicate ± SEM; semi-transparent lines indicate linear fits to the data. (**e**) The correlation between pupil size and sequence as a function of time. Note, we did not analyze pupillometry data for micro temporal context due to the confounding effect of differential foveal luminance between repeat and alternate stimuli. (**f**) Classification accuracy of stimuli presented on different sides of task-related planes as a function of time, for each of the eight sequences, from re-analysis of previously published EEG data (*Rideaux, 2024*). (**g**) Same as (**f**), but for the task-unrelated plane. The inset in (**f**) shows the EEG sensors included in the analysis (blue dots). Black rectangles indicate the timing of stimulus presentations (solid: target stimulus, dashed: previous and subsequent stimuli). Shaded regions indicate ± SEM. Horizontal bars indicate cluster-corrected periods of significant relationships between (**e**) pupil size or (**f, g**) classification accuracy and sequence order.

The online version of this article includes the following figure supplement(s) for figure 4:

**Figure supplement 1.** Meso-scale temporal context effects in Experiment 2.

**Figure supplement 2.** Removal of general meso temporal dependencies in EEG responses.

the pupils dilate in what has been interpreted as an arousal response (*Mazancieux et al., 2023*; *Preuschoff et al., 2011*). We found that participants' pupils dilated more in response to unexpected sequences (*Figure 4d*), indicating that meso temporal context establishes sensory predictions. Another possible explanation why we did not detect an effect on recall precision is that we lacked statistical power. The micro analyses split data between two conditions (repeat and alternate), whereas in the meso analysis, data is split between eight (sequences). Indeed, in our second

experiment, we reproduced the response time and task accuracy effects of meso temporal context and further found that unexpected sequences were recalled more precisely (*Figure 4—figure supplement 1*).

To further test whether meso temporal context effects were qualitatively different from those at the micro scale, we separated the EEG classification accuracies (*Figure 2e*) into the eight sequence categories and corrected for shuffled accuracy (*Figure 4—figure supplement 2*). We found a strikingly similar pattern of results to those observed at the micro scale: there was increased classification accuracy for unexpected sequences from ~60 to 100ms along both task-related and unrelated reference planes (*Figure 4h and g*). This finding provides further evidence of a passive mechanism that suppresses feedforward sensory activity elicited by expected stimuli and leads to more precise recall of those that are unexpected. That the effect was reproduced when repeat and alternate stimuli were matched also shows that it cannot be explained by sensory adaptation (*Rideaux et al., 2023*).

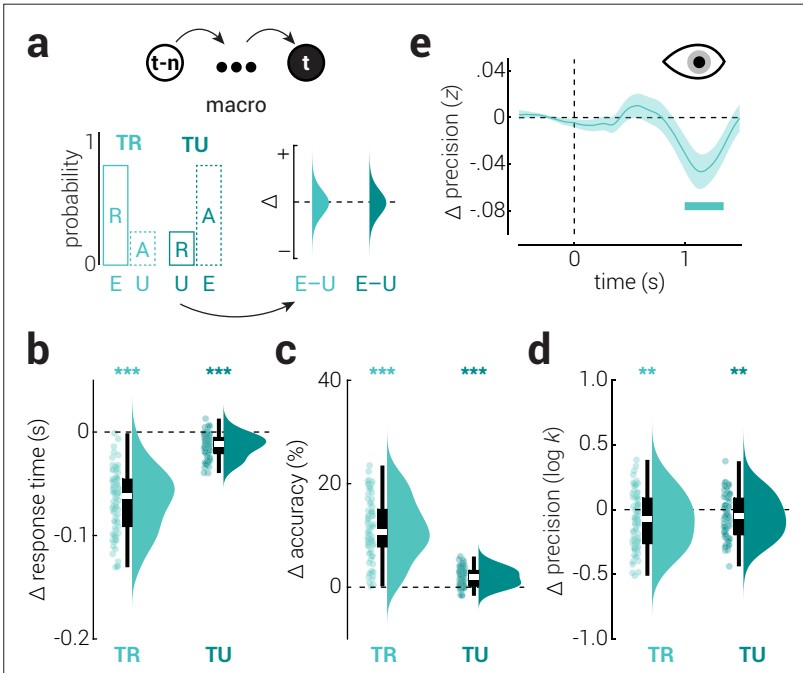

**Figure 5.** The influence of macro temporal context on visual processing. (**a**) Repeat (**R**) and alternate (**A**) stimuli were presented with unequal probabilities (counterbalanced across participants) along task-related and unrelated reference planes. The difference in performance between relatively frequent (expected; **E**) and infrequent (unexpected; **U**) stimuli was calculated to assess the influence of macro temporal context on visual processing. (**b–d**) The difference (expected – unexpected) in (**b**) response time, (**c**) task accuracy, and (**d**) reproduction precision between expected and unexpected stimuli, along task-related (TR) and unrelated (TU) planes. Note that we observed task accuracy differences associated with macro temporal context across the task-unrelated plane. In the micro and meso analyses, the expected outcome along the task-unrelated plane was never in conflict with that in the task-related plane. For example, if a stimulus is expected on the same (right) side of the display (repeat; task-related plane), then regardless of whether the stimulus appeared above or below fixation, there was a quadrant of the display that would satisfy both the task-related and unrelated expectations (bottom- or top-right). This is because the expectation was either for repeat stimuli (micro) or balanced between repeat and alternate stimuli (meso). By contrast, in the macro condition, where stimuli could be expected to alternate, this produced trials on which the task-related and unrelated expectations conflicted, such that there was no location that could satisfy both. Accuracy along the task-unrelated plane was reduced in these instances where there was conflict. Asterisks indicate significant differences (**p<0.01, ***p<0.001). (**e**) The difference (expected – unexpected) in pupil size as a function of time from stimulus presentation, along the task-related (TR) plane. Shaded region indicates ± SEM and the horizontal bar indicates a cluster-corrected period of significant difference.

The online version of this article includes the following figure supplement(s) for figure 5:

**Figure supplement 1.** Macro-scale temporal context effects in Experiment 3.

## Macro-scale temporal context

Events within micro and meso temporal contexts occur within the limits of short-term memory. Most events that occur further into the past cannot be recalled; yet, they continue to influence perception (*Schapiro and Turk-Browne, 2015*; *Simoncelli and Olshausen, 2001*). To test the influence of macro temporal context on perception, we ran a second experiment in which the probability of stimulus presentation was biased towards repeats or alternates along task-related and unrelated reference planes (*Figure 1d*, Experiment 2). The biases were counterbalanced across participants such that we could compare performance between relatively frequent (expected) and infrequent (unexpected) events, while controlling for differences between repeat and alternate stimuli (*Figure 5a*). Consistent with micro and meso scale effects, we found that expected events were responded to faster and more accurately, while unexpected events were reproduced more precisely, across both task-related (speed: $z_{78}=7.67$, p=$1.68e^{-14}$, r=0.87; task accuracy: $z_{73}=7.42$, p=$1.13e^{-13}$, r=0.87; precision: $z_{79}=3.26$, p=0.001, r=0.37) and unrelated planes (speed: $z_{71}=6.69$, p=$2.19e^{-11}$, r=0.79; task accuracy: $z_{78}=6.76$, p=$1.32e^{-11}$, r=0.77; precision: $z_{75}=2.72$, p=0.007, r=0.31; *Figure 5b–d*). This effect was larger along the task-related plane for response time and task accuracy, but not precision (speed: $z_{71}=7.24$, p=$4.42e^{-13}$, r=0.86; task accuracy: $z_{78}=7.27$, p=$3.52e^{-13}$, r=0.82; precision: $z_{75}=0.93$, p=0.350, r=0.11). Pupillometry analysis further revealed that participants' pupils dilated more in response to unexpected stimuli ~1 s after stimulus onset (*Figure 5e*).

In line with previous work, stimuli in our experiment were displayed until participants responded. To test how variable presentation duration influenced the results, we ran a third experiment that was the same as the second, except with a fixed stimulus duration (200ms). Experiment 3 replicated both the behavioral and pupillometry results from Experiment 2 (*Figure 5—figure supplement 1*), indicating that the effects observed were not related to variable stimulus duration.

# Discussion

Evidence from multiple, separate, fields of research shows that humans are sensitive to the temporal context in which sensory stimuli are embedded across a broad range of time scales. Here we used a novel experimental paradigm to characterize and compare how past sensory events occurring at different temporal scales shape perception of the present. By combining behavioral, neuroimaging, and physiological approaches to assess the influence of temporal context at the micro, meso, and macro scale, we identified new properties of these phenomena in isolation and further revealed two distinct adaptive mechanisms that unify them.

## Micro-scale temporal context

At the micro scale, stimuli were responded to faster and more accurately when preceded by those that were similar, but recalled more precisely when preceded by those that were different. This may reflect a general expectation that objects in the environment are more likely to stay the same than to change (*Fischer and Whitney, 2014*). Under this interpretation, expected events are rapidly responded to while unexpected events are precisely encoded. As will be discussed shortly, this interpretation resonates with the influence of temporal context at longer scales.

Analysis of EEG recordings showed that during the initial stage of stimulus processing in the cortex (60ms after onset), alternating stimuli were represented both (20ms) earlier and more robustly than those that repeat. This finding provides a neural correlate of the improved recall precision for these stimuli and, consistent with recent psychophysical work (*Rideaux et al., 2024*), suggests that unexpected (alternating) events are prioritized in terms of both encoding fidelity and processing speed. Given that unexpected stimuli were responded to more slowly than expected stimuli, despite benefitting from sensory processing priority, this may indicate that the latter is supported by anticipatory motoric activity (*Jentzsch and Sommer, 2002*).

Effects on binary judgements associated with micro temporal context, such as those described above, have been referred to as first-order sequential dependencies (*Bertelson, 1961*). By contrast, those associated with continuous report tasks are typically referred to as serial dependence (*Cicchini et al., 2014*; *Fischer and Whitney, 2014*). We examined behavioral responses for continuous effects and found the stereotypical pattern of response biases associated with serial dependence, that is attraction towards the preceding stimulus.

Our corresponding EEG analyses revealed better decoding accuracy and precision for stimuli preceded by those that were different and a bias away from the previous stimulus. These results are consistent with findings that alternating stimuli are recalled more precisely. Further, while the repulsive pattern of biases is inconsistent with the observed behavioral attractive biases, it is consistent with recent work on serial dependence indicating an initial period of repulsion, followed by an attractive bias during the response period (*Luo et al., 2025*; *Sheehan and Serences, 2022*; *Fischer et al., 2024*). These findings indicate that serial dependence and first-order sequential dependencies can be explained by the same underlying principle. Behavioral and neural recordings show a predisposition for expected outcomes, which manifest as faster, biased, behavioral responses. By contrast, unexpected outcomes are encoded earlier and more precisely, leading to increased recall fidelity for these events.

## Meso-scale temporal context

At the meso scale, consistent with previous work (*Kirby, 1976*; *Remington, 1969*), we found that short sequences of stimuli were responded to faster and more accurately when they were expected. Extending these findings, and matching the influence of micro-scale temporal context, we also found that responses were more precise for unexpected sequences (in Experiment 2, and non-significantly in Experiment 1). Pupillometry analyses confirmed that participants were surprised by stimuli that were categorized as unexpected, and EEG analyses revealed a strikingly similar pattern of results as was found at the micro scale. In particular, unexpected stimuli were encoded earlier and more robustly. The similarity between the influence at the micro and meso scale is particularly remarkable given that the influence of micro temporal context was controlled for in the meso-scale analyses.

Theoretical and empirical work has implicated both motor and sensory priming in the modulation of response time and accuracy associated with higher-order sequential dependencies (*Jentzsch and Sommer, 2002*; *Gao et al., 2009*; *Galluzzi et al., 2022*). Decoupling the motoric component of experimental designs used to probe the phenomenon has been an obstacle to assessing how sensory processing is influenced by meso-scale temporal context. In the EEG experiment that was re-analyzed here, participants viewed serially presented stimuli and were tasked with reporting the number of targets after 30 presentations. Thus, there were no motor actions generated during the recording period used to perform the analysis, and our results can only be attributed to differences in visual processing between sequences. Indeed, the timing of the improved encoding of unexpected sequences, like that for micro temporal context, can only be accounted for by a mechanism that prioritizes the encoding speed and fidelity of stimuli during the initial feedforward cascade of sensory processing.

## Macro temporal context

Results from our second experiment, in which biases in the probability of repeat and alternate stimuli were introduced, showed that responses are faster and more accurate for expected stimuli, and that recall precision is better for unexpected stimuli. Pupillometry analysis confirmed that participants were surprised by unexpected stimuli. We replicated these results in a third experiment, demonstrating that the same outcome for stimuli of fixed duration. These findings parallel those observed at the micro and meso scales and point to two canonical adaptive mechanisms that operate across a wide range of temporal contexts to support a common sensory outcome.

## The role of attention

At each scale, we found that attention (operationalized through task-relatedness) amplified the influence of temporal context for response time and task accuracy, but not precision. These findings suggest that perception and behavior are modulated by expectation through two distinct mechanisms. One mechanism is sensitive to top-down goals (attention) and supports rapid responses to expected stimuli, while the other mechanism automatically detects mismatches between expected and actual outcomes and prioritizes the encoding speed and fidelity of unexpected stimuli. In line with this, some studies have reported interactions between attention and predictive mechanisms (*Auksztulewicz and Friston, 2015*; *Smout et al., 2019*), while others have shown that oddball event-related potentials can be detected in coma patients (*Guérit et al., 1999*).

Consistent with previous work (*Rideaux, 2024*), we found that the influence of attention on the neural representation only emerged after ~200 ms. By contrast, we found that effects of expectation, for both micro- and meso-scale temporal context, were present during the earliest moments of cortical sensory processing, that is during the feedforward cascade. These findings suggest that predictive mechanisms are operational during feedforward processing, while attention modulates sensory processing at a later stage.

## Predictive coding

Predictive coding theories argue that *prediction errors* are generated when bottom-up sensory inputs deviate from top-down expectations (*Friston, 2005*). There is broad consensus that prediction errors are associated with increased neural activation (*Kok et al., 2012*; *Alink et al., 2010*; *den Ouden et al., 2012*; *Meyer and Olson, 2011*; *Richter et al., 2018*; *Todorovic et al., 2011*), which is typically observed during the initial processing cascade in sensory cortices associated with the unexpected stimulus (*Tang et al., 2018*; *Tang et al., 2023*), but has also been reported in subcortical regions (*Mazancieux et al., 2023*). By contrast, whether this modulation of neural activation results from 'dampening' or 'sharpening' of the neurons tuned to expected features remains highly contested (*Friston, 2005*; *Kok et al., 2012*; *de Lange et al., 2018*; *Richter et al., 2022*; *Hu et al., 2025*). Whether through increased selectivity (sharpening) or reduced sensitivity (dampening), both accounts predict attenuated activation in response to expected events. However, according to the sharpening model, expected events are represented more precisely, while the dampening model posits the opposite, that is unexpected events are more precisely represented. At every scale of temporal context and across behavioral and neural recordings, we consistently found that unexpected events were encoded more precisely than expected events. These findings provide compelling evidence that predictive mechanisms reduce the sensitivity of neurons tuned to expected features, likely in order to reduce metabolic energy expenditure (*Friston, 2005*), rather than increasing their selectivity.

The expectation of stimuli shaped their neural representation in a robust and striking manner. In particular, the peak in classification accuracy from 50 to 150ms following stimulus onset, which represents the initial stage of (feedforward) cortical representation, was almost entirely accounted for by stimulus expectation such that expected stimuli were only minimally represented during this period. This modulation cannot be attributed to adaptation, as we found the same result at the meso scale, where the change in location between subsequent stimuli was matched between conditions. There was relatively weak evidence for a difference between the neural representation of expected and unexpected stimuli following this initial peak. Given that unexpected stimuli were recalled more precisely in the behavioral task, this provides further support to the notion that recall of these stimuli primarily reflects neural activity during this initial stage of sensory processing; however, more work is needed to understand how this relationship changes with increasing stimulus complexity. These findings provide empirical evidence that predictive mechanisms operate to reduce the feedforward sensory activity associated with expected stimuli in a manner that reduces their representational fidelity (i.e. dampening/cancellation).

Prioritizing the speed and fidelity with which unexpected events are encoded likely serves to rapidly and precisely update internal predictive models when there is a mismatch between predictions and incoming sensory information. By contrast, expected events may benefit from anticipatory motoric activation, which facilitates rapid behavioral responses and do not require encoding prioritization.

## Limitations and future directions

One potential limitation of the current study is the categorization of temporal scales according to events, independent of the influence of event duration. While this simplification of time supports comparison between different phenomena associated with each scale (e.g. serial dependence, sequential dependencies, statistical learning), future work could investigate the role of duration to provide a more comprehensive understanding of the mechanisms identified in the current study. Related to this, while the temporal scales applied here conveniently categorized known sensory phenomena and partially correspond to iconic-, short-, and long-term memory, there are alternative ways of delineating time. For example, temporal scales could be defined simply as short- and long-term (*Xie et al., 2025*). However, this could obscure meaningful differences between phenomena

associated with sensory persistence and short-term memory, or qualitative differences in the way that short-sequences of events are processed.

Another limitation of the current study is that the EEG recordings were collected in a separate experiment to the behavioral and pupillometry data. The stimuli and task were similar between experiments, but not identical. For example, the EEG experiment employed colored arc stimuli presented at a constant rate of ~3.3 Hz and participants were tasked with counting the number of stimuli presented at a target location. By contrast, in the behavioral experiment, participants viewed white blobs presented at an average rate of ~2.8 Hz and performed a binary spatial task coupled with an infrequent reproduction task. An advantage of this was that the sensory responses to stimuli in the EEG recordings were not conflated with motor responses; however, future work combining these measures in the same experiment would serve as a validation for the current results.

## Conclusion

Past sensory information shapes perception of the present. Multiple research areas have emerged to investigate the isolated influence of temporal context at different scales. In a single experimental paradigm, we compared the influence of past stimuli at micro, meso, and macro temporal scales and found two common rules that characterized the relationship between the past and present across all ranges. While the neural mechanisms that implement the influence of temporal context across different scales may vary (e.g. tonic inhibition/excitation, synaptic weight), they appear to promote two unifying outcomes: (1) rapid motor responses to expected events and (2) prioritized encoding speed and fidelity of unexpected events.

## Methods

### Participants

Eighty neurotypical human adults participated in Experiments 1 and 2 (mean ± standard deviation age, Experiment 1: 20.9±2.9 years, 14 males, 65 females, 1 non-binary, Experiment 2: 21.0±5.8 years, 18 males, 62 females), and 40 neurotypical human adults participated in Experiment 3 (24.7±8.3 years, 10 males, 30 females). The data of one participant was omitted from Experiment 1 because they were unable to complete the experiment. Sample size was informed by previous studies using similar psychophysical methods (*Rideaux et al., 2024*). Participants were recruited from The University of Sydney and had normal or corrected-to-normal vision (assessed using a standard Snellen eye chart). All participants were naïve to the aims of the experiment and gave informed written consent. The experiment was approved by The University of Sydney Human Research Ethics Committee (2023/HE000072).

### Apparatus

The experiment was conducted in a dark acoustically shielded room. The stimuli were presented on a 41.5-inch ASUS ROG Swift OLED monitor with 3840x2, 160 resolution and a refresh rate of 120 Hz. Viewing distance was maintained at 1 m using a chin and head rest. Stimuli were generated in MATLAB v2020a (The MathWorks, Inc, Matick, MA) using Psychophysics Toolbox (*Brainard, 1997*; *Pelli, 1997*) v3.0.18.13 (see http://psychtoolbox.org/). Gaze direction and pupil diameter were recorded monocularly (right eye) at 1 kHz using an EyeLink 1000 (SR Research Ltd., Ontario, Canada).

### Stimuli, task, and procedure

The stimuli comprised white Gaussian blobs (standard deviation = 0.3°, contrast = 0.5) positioned 4° of visual angle from fixation (location randomly selected between 0 and 360°) on a black background. A centrally positioned white fixation dot (radius = 0.25°) was presented to reduce eye movements. Stimuli were presented until the participant responded. Half the participants were instructed to use the 'z' and '/' keys to indicate whether the stimulus appeared on the left or right of fixation, and the other half used the 'b' and 't' keys to indicate whether it appeared above or below fixation; the keys were selected to map the relative positions on the keyboard to the orientation of the task. Participants performed 15–20 blocks of 100 trials (~90 min), receiving feedback on their response time and task accuracy at the end of each block. On 10% of randomly selected trials, after participants responded, the target blob was replaced by a green blob at a random location (same size as target blob), and

participants were instructed to use the response keys to rotate the location of the green blob around to the location of the previous target blob and press the 'spacebar' key to confirm their response.

Experiment 2 was the same as Experiment 1, except that biases were introduced to the location of the target blob. In particular, sequentially presented target stimuli were more likely to be presented on the same side (75%; repeat) than on the other side (25%; alternate), or vice versa, along each of the task reference planes (left/right and above/below). Thus, a 2 (task reference plane)×2 (left/right bias)×2 (above/below bias) design was used and the eight conditions were counterbalanced across participants. On each trial, based on the location of the target on the previous trial and the unique bias condition, the quadrant of the next target was probabilistically determined and then the location was randomly selected within the quadrant. Critically, the biases manipulated the probability of repeat and alternate stimuli, not the spatial location (e.g., left or right). Experiment 3 was the same as Experiment 2, but stimuli were presented for a fixed duration (0.2 s). This duration was selected because it was below the normal range of response times and was intended to allow the stimulus to be presented for the full duration before participants responded.

## Eye tracking

Pupil size recordings were epoched to between –0.5 and 1.5 s around target stimulus presentation. Pupillometry data were preprocessed by removing blinks (0.1 s buffer window), removing outliers (median absolute deviation >2.5), interpolated, bandpass filtered with a first-order Butterworth filter (highpass = 15.6 Hz, lowpass = 500 Hz), $z$-scored, baselined to the 0.5 s period before stimulus presentation, and downsampled to 125 Hz, respectively. Due to hardware issues, pupil data were not collected from one participant in Experiment 1, two in Experiment 2, and three in Experiment 3.

## Behavioral analyses

Trials were grouped according to whether stimuli repeated (the previous stimulus was presented on the same side) or alternated (the previous stimulus was presented on the opposite side) according to three reference planes: task-related, task-unrelated, and stimulus-related. The task-related plane mapped onto the participants' task condition (left-right or above-below) and the task-unrelated plane was orthogonal to the task-related plane. For example, if a participant was assigned the left-right task condition, the task-related plane was defined by the vertical meridian and the task-unrelated plane was defined by the horizontal meridian. The stimulus-related plane was defined by the line bisecting the fixation point at an angle orthogonal to the stimulus location on the previous trial, and thus changed over time.

For micro analyses, the difference in responses to repeat and alternate stimuli was compared. For meso analyses, trials were further sorted into eight pairs of unique sequences, based on the (repeat/alternate) identity of the previous five trials (*Figure 4a*). For macro analyses in Experiment 2 and 3, repeat and alternate trials were grouped according to whether they were expected (75% probability) or unexpected (25% probability), based on participants' experimental condition. For all analyses, sequences that were disrupted by a reproduction task or block advancement were omitted. For each participant, after removal of trials where response time exceeded 2 s, median response times and average task accuracy were calculated.

From the reproduction task, we sought to estimate participants' recall precision. It is likely that on some trials, participants failed to encode the target and were forced to make a response guess. To isolate the recall precision from guess responses, we used mixture modeling to estimate the precision and guess rate of reproduction responses, based on the concentration (κ) and height of von Mises and uniform distributions, respectively (*Bays et al., 2009*). The κ parameter of the von Mises distribution reflects its width, which indicates the clustering of responses around a common location. For the serial dependency analysis, we calculated bias and precision by binning reproduction responses according to the angular offset between the current and previous target and then performing mixture modeling to estimate the mean (bias) and κ (precision) parameters of the von Mises distribution. Precision values were positively skewed across participants, so a logarithmic transformation was applied to normalize their distribution ($\log_{10}κ$). For serial dependence analyses, the location ($\mu$) of the von Mises distribution was used to estimate response bias.

## EEG

EEG data from a previously published dataset were re-analyzed (*Rideaux, 2024*). The stimuli comprised colored arcs (inner ring, 0.25°; outer ring, 4.25°) extending 45° polar angle on a mid-gray background. Trials consisted of 30 arc stimuli (location and color randomly selected between 0° and 360°) presented for 0.2 s each, separated by a blank 0.1 s inter-stimulus interval. Participants were instructed to indicate the number of target stimuli that were presented during the preceding trial. At the beginning of each block, target stimuli were identified as either those appearing at a defined location or color, counterbalanced across the session. EEG data from blocks in which the target was defined by spatial location were included in micro and meso linear discriminant analyses; all data were included in the sequential dependency forward encoding analysis.

The EEG recordings were digitized at 1024 Hz sampling rate with a 24-bit A/D conversion. The 64 active scalp Ag/AgCl electrodes were arranged according to the international standard 10–20 system for electrode placement (*Oostenveld and Praamstra, 2001*) using a nylon head cap and with an online reference of FCz. Offline EEG pre-processing was performed using EEGLAB v2021.1 (*Delorme and Makeig, 2004*). The data were initially subjected to a 0.1 Hz high-pass filter to remove slow base-line drifts and a 45 Hz low-pass filter to remove high-frequency noise/artifacts before being downsampled to 512 Hz. Data were then re-referenced to the common average before being epoched into segments around each stimulus (–0.125 s to 0.5 s from the stimulus onset). We only included the parietal, parietal-occipital, and occipital sensors in the analyses to (*i*) reduce the influence of signals produced by eye movements, blinks, and non-sensory cortices, (*ii*) for consistency with similar previous decoding studies (*Rideaux, 2024*; *Buhmann et al., 2024*; *Rideaux et al., 2025*), and (*iii*) to improve decoding accuracy by restricting sensors to those that carried spatial position information (*Figure 2— figure supplement 1*).

## Neural decoding

For micro and meso analyses, EEG recordings (epochs) were grouped according to which side of the task-related or unrelated reference plane they were presented. As the task in the EEG experiment was to detect stimuli presented at a specific location, rather than perform a binary judgement (e.g. left or right of fixation), we constructed task boundaries based on the spatial coordinates that were indicative to the target detection task. Thus, the task-related reference plane was defined by the line bisecting the fixation point at an angle orthogonal to the target location, because this plane was most informative of whether a stimulus was a target. By contrast, the task-unrelated plane was the line orthogonal to the task-related line, as this plane was uninformative of whether a stimulus was a target. Tenfold cross-validation linear discriminant analysis was performed with the MATLAB *classify* function (discriminant function, diaglinear) to calculate the two-way classification accuracy separately at each time point and reference plane. Classification accuracy was then separately assessed for repeat and alternate trials (micro analysis) and the eight pairs of five-event sequences (meso analysis).

As shown in *Figure 2—figure supplement 2*, we found that there were differences in classification accuracy for repeat and alternate stimuli in the EEG data, even when stimulus labels were shuffled. This is likely due to temporal autocorrelations within the EEG data that are unrelated to the decoded stimulus dimension. This signal causes the decoder to classify temporally proximal stimuli as the same class, leading to a bias towards repeat classification. For example, in general, the EEG signal during trial one will be more similar to that during trial two than during trial ten, because of low-frequency trends in the recordings. If the decoder has been trained to classify the signal associated with trial one as a leftward stimulus, then it will be more likely to classify trial two as a leftward stimulus too. These autocorrelations are unrelated to stimulus features; thus, to isolate the influence of stimulus-specific temporal context, we subtracted the accuracy produced by shuffling the stimulus labels from the unshuffled accuracy. As a further test of this explanation, we performed the same classification analysis on the data after applying a high-pass filter of 0.7 Hz.

For serial dependency analyses, we used an inverted modeling approach to reconstruct either the location or Δ location (angular distance from previous stimulus) of the stimuli (*Brouwer and Heeger, 2011*); note, the results of Δ location analyses are shown in *Figure 2—figure supplement 1*. A theoretical (forward) model was nominated that described the measured activity in the EEG sensors given the location/Δ location of the presented stimulus. The forward model was then used to obtain the

inverse model that described the transformation from EEG sensor activity to stimulus location/Δ location. The forward and inverse models were obtained using a ten-fold cross-validation approach.

Similar to previous work (**Rideaux et al., 2023**; **Brouwer and Heeger, 2009**; **Harrison et al., 2023**), the forward model comprised five hypothetical channels, with evenly distributed idealized location/Δ location preferences between 0° and 360°. Each channel consisted of a half-wave rectified sinusoid raised to the fifth power. The channels were arranged such that a tuning curve of any location/Δ location preference could be expressed as a weighted sum of the five channels. The observed EEG activity for each presentation could be described by the following linear model:

$$\mathbf{B} = \mathbf{WC} + \mathbf{E} \tag{1}$$

where $\mathbf{B}$ indicates the ($m$ sensors × $n$ presentations) EEG data, $\mathbf{W}$ is a weight matrix ($m$ sensors × 5 channels) that describes the transformation from EEG activity to stimulus location/Δ location, $\mathbf{C}$ denotes the hypothesized channel activities (5 channels × $n$ presentations), and $\mathbf{E}$ indicates the residual errors.

To compute the inverse model, we estimated the weights that, when applied to the data, would reconstruct the underlying channel activities with the least error. In line with previous magnetoencephalography work (**Kok et al., 2017**; **Mostert et al., 2015**), when computing the inverse model, we deviated from the forward model proposed by **Brouwer and Heeger, 2009** by taking the noise covariance into account to optimize it for EEG data, given the high correlations between neighboring sensors. We then estimated the weights that, when applied to the data, would reconstruct the underlying channel activities with the least error. Specifically, $\mathbf{B}$ and $\mathbf{C}$ were demeaned such that their average over presentations equaled zero for each sensor and channel, respectively. The inverse model was then estimated using a subset of data selected through cross-fold validation. The hypothetical responses of each of the five channels were calculated from the training data, resulting in the response row vector $c_{train,i}$ of length $n_{train}$ presentations for each channel $i$. The weights on the sensors $\mathbf{w}_i$ were then obtained through least squares estimation for each channel:

$$\mathbf{w}_i = \boldsymbol{B}_{train}\boldsymbol{c}_{train,i}^{\mathrm{T}} \left( \mathbf{c}_{train,i}\mathbf{c}_{train,i}^{\mathrm{T}} \right)^{-1} \tag{2}$$

where $B_{train}$ indicates the ($m$ sensors × $n_{train}$ presentations) training EEG data. Subsequently, the optimal spatial filter $v_i$ to recover the activity of the $i$th channel was obtained as follows **Mostert et al., 2015**:

$$\mathbf{v}_i = \frac{\tilde{\Sigma}_i^{-1}\mathbf{w}_i}{\mathbf{w}_i^{\mathrm{T}}\tilde{\Sigma}_i^{-1}\mathbf{w}_i} \tag{3}$$

where $\tilde{\Sigma}_i$ is the regularized covariance matrix for channel $i$. Incorporating the noise covariance in the filter estimation leads to the suppression of noise that arises from correlations between sensors. The noise covariance was estimated as follows:

$$\hat{\Sigma}_i = \frac{1}{n_{\mathrm{train}} - 1} \varepsilon_i \varepsilon_i^{\mathrm{T}} \tag{4}$$

$$\varepsilon_i = \mathbf{B}_{train} - \mathbf{w}_i\mathbf{c}_{train,i} \tag{5}$$

where $n_{train}$ is the number of training presentations. For optimal noise suppression, we improved this estimation by means of regularization by shrinkage using the analytically determined optimal shrinkage parameter (**Mostert et al., 2015**), yielding the regularized covariance matrix $\tilde{\Sigma}_i$.

For each presentation, we decoded location and Δ location by converting the channel responses to polar form:

$$z = \mathbf{c} \cdot e^{2i\varphi} \tag{6}$$

and calculating the estimated angle:

$$\hat{\theta} = \frac{\arg(z)}{2} \tag{7}$$

where $c$ is a vector of channel responses and $\varphi$ is the vector of angles at which the channels peak. Decoding accuracy, which represents the similarity of the decoded location/colour to the presented location and Δ location (*Kok et al., 2017*), was expressed by projecting the mean resultant (presentations averaged across 12 evenly distributed Δ location bins) of the difference between decoded and arc location onto a vector with 0°:

$$\hat{r}_\theta = \text{Re}[\bar{R}], \quad \bar{R} = \frac{1}{n}\sum_{i=1}^{n} \exp\left(\text{i}(\hat{\theta}_j - \theta)\right) \tag{8}$$

Precision was estimated by calculating the angular deviation (*Zar, 1999*) of the decoded orientations within each orientation bin:

$$\hat{\sigma}_\theta = \sqrt{2(1 - |\bar{R}|)} \tag{9}$$

and normalized, such that values ranged from 0 to 1, where 0 indicates a uniform distribution of decoded orientations across all orientations (i.e. chance-level decoding) and 1 represents perfect consensus among decoded orientations:

$$\hat{p}_\theta = 1 - \frac{2\hat{\sigma}_\theta}{\sqrt{2}} \tag{10}$$

Bias was estimated by computing the circular mean of angular difference between the decoded and presented orientation:

$$\hat{b}_\theta = \arg\left(\bar{R}\right) \tag{11}$$

As shown in *Figure 3—figure supplement 1*, we found the same general temporal dependencies in the decoding accuracy computed using inverted encoding that were found using linear discriminant classification. However, as a baseline correction would not have been appropriate or effective for the parameters decoded with this approach, we instead used a high-pass filter of 0.7 Hz to remove the confound, while being cautious about interpreting the timing of effects produced by this analysis due to the temporal smear introduced by the filter.

## Statistical analyses

Statistical analyses were performed in MATLAB v2020a and CircStat Toolbox v1.12.0.0 (*Berens, 2009*). Prior to statistical testing of behavioral estimates, within-participant average values of response time, task accuracy, and precision estimates >1.5 times the interquartile range of the group distribution were removed. Non-parametric inferential tests (Wilcoxon signed rank test *Wilcoxon and Johnson, 1992* and Friedman's ANOVA *Friedman, 1937*) were used to assess the significance of paired differences and main effects. For Wilcoxon signed rank tests, the rank-biserial correlation (r) was calculated as an estimate of effect size, where 0.1, 0.3, and 0.5 indicate small, medium, and large effects, respectively (*Cureton, 1956*). For Friedman's ANOVA tests, Kendall's W was calculated as an estimate of effect size, where 0.1, 0.3, and 0.5 indicate small, medium, and large effects, respectively (*Kendall, 1948*). Levene's test of homogeneity was used to compare the difference in variance between conditions (*Levene, 1960*). For pupillometry and linear discriminant analyses, one-dimensional cluster correction was applied to remove spurious significant differences (*Pernet et al., 2015*). For inverted modeling analyses of location and Δ location as a function of Δ location and time, a two-dimensional circular-linear cluster correction was applied. First, at each Δ location and/or time point, the effect size of the dependent variable (e.g., decoding accuracy) was calculated. Next, we calculated the summed value of these statistics (separately for positive and negative values) within contiguous featural and/or temporal clusters of significant values. We then simulated the null distribution of the maximum summed cluster values using permutation (n=5000) of the sign or condition labels, for single and paired *t*-test comparisons, respectively, from which we derived the 95% percentile threshold value. Clusters identified in the data with a summed effect size value less than the threshold were considered spurious and removed. Prior to cluster analysis, we smoothed the decoding accuracy estimates in the inverted modeling analyses along the feature dimension using a uniform kernel (size, 4).

## Acknowledgements

We thank Arnaldo Bisbal, Leon Zhong, and Cecelia Chenh for assistance with data collection. We also thank Dr William Harrison for their feedback on an earlier version of the manuscript. This work was supported by Australian Research Council (ARC) Discovery Early Career Researcher Awards awarded to RR (DE210100790) a National Health and Medical Research Council (NHMRC; Australia) Investigator Grant (2026318).

## Additional information

### Funding

| Funder | Grant reference number | Author |
|---|---|---|
| Australian Research Council | DE210100790 | Reuben Rideaux |
| National Health and Medical Research Council | 2026318 | Reuben Rideaux |

The funders had no role in study design, data collection and interpretation, or the decision to submit the work for publication.

### Author contributions

Kacie Lee, Conceptualization, Data curation, Writing – review and editing; Reuben Rideaux, Conceptualization, Resources, Formal analysis, Supervision, Funding acquisition, Investigation, Visualization, Methodology, Writing – original draft, Project administration, Writing – review and editing

### Author ORCIDs

Reuben Rideaux (iD) https://orcid.org/0000-0001-8416-005X

### Ethics

Human subjects: All participants were naïve to the aims of the experiment and gave informed written consent. The experiment was approved by The University of Sydney Human Research Ethics Committee (2023/HE000072).

Reviewer #1 (Public review): https://doi.org/10.7554/eLife.106614.3.sa1
Reviewer #2 (Public review): https://doi.org/10.7554/eLife.106614.3.sa2
Author response https://doi.org/10.7554/eLife.106614.3.sa3

## Additional files

### Supplementary files

MDAR checklist

### Data availability

The data and analysis code generated in this study have been deposited in the following OSF database: https://osf.io/9qfdk/.

The following dataset was generated:

| Author(s) | Year | Dataset title | Dataset URL | Database and Identifier |
|---|---|---|---|---|
| Rideaux R | 2024 | The influence of temporal context on vision over multiple time scales | https://doi.org/10.17605/OSF.IO/9QFDK | Open Science Framework, 10.17605/OSF.IO/9QFDK |

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
