## [Editor Report · eLife Assessment]

This **fundamental** study shows how past experiences shape perception across short, medium, and long time scales, using a single behavioural paradigm and reanalysed EEG data. It provides **convincing** evidence for two processes across all scales: an attention-dependent mechanism that speeds responses to expected events, and an attention-independent mechanism where expected events are encoded less precisely, consistent with feedforward dampening. The work offers a unifying account of temporal context effects, though stronger brain-behaviour links, integration with serial dependence attraction and repulsion models, and extension to other timescale definitions would further strengthen the contribution.

---

## [Referee Report · Reviewer #1 (Public review)]

Summary:

This paper addresses an important and topical issue: how temporal context - at various time scales - affects various psychophysical measures, including reaction times, accuracy and localization. It offers interesting insights, with separate mechanisms for different phenomena, which are well discussed.

Strengths:

The paradigm used is original and effective. The analyses are rigorous.

Comments on revised version:

I think the authors have dealt adequately with my issues, none of which were fundamental.

---

## [Referee Report · Reviewer #2 (Public review)]

Summary:

This study investigates the influence of prior stimuli over multiple time scales in a position discrimination task, using pupillometry data and a reanalysis of EEG data from an existing dataset. The authors report consistent history-dependent effects across task-related, task-unrelated, and stimulus-related dimensions, observed across different time scales. These effects are interpreted as reflecting a unified mechanism operating at multiple temporal levels, framed within predictive coding theory.

Strengths:

The authors have done a good job in their revision, clarifying important points and stating the limitations of the study clearly.

I also think they made a valid effort to address and correct issues arising from the temporal dependency confound, although I still wonder whether the best approach would have been to design an experiment in a way that avoided this confound in the first place.

Overall, this is a substantially improved version, and I particularly appreciate the clarification and correction regarding the direction of the bias in the EEG data (repulsive rather than attractive).

Weaknesses:

These are now relatively minor points.

I believe this latter aspect, the repulsive bias, may deserve further discussion, especially in relation to their behavioral findings and, in particular, to earlier work proposing multi-stage frameworks of serial dependence, where low-level repulsion interacts with attractive biases at higher-level stages (Fritsche et al., 2020; Pascucci et al., 2019; Sheehan & Serences, 2022). The authors may also consider to cite some key reviews on serial dependence that discuss both repulsion and attraction in forced-choice and reproduction tasks (Manassi et al., 2023; Pascucci et al., 2023).

Related to this, after finding the opposite pattern, is the sentence in line 472-473 ("Further, we found an attractive...") and the related argument still valid?

Regarding my earlier point about former line 197 and Figure 3b,c: what I noticed-similar to the patterns reported in the studies I referenced-is that the data cannot be simply described as showing faster and more accurate responses for small deltas. Responses also appear faster and more accurate for very large deltas, with performance being worse in between. Indeed, as the authors state: "The peak in precision for large Deltas locations is consistent with alternate events being encoded more precisely, while the peak for small offsets may be explained by the attractive bias towards the previous target." I wonder whether it is necessary, or unequivocally supported by the data, to hypothesize two separate mechanisms here. An alternative could be interference effects between consecutive stimuli that are neither identical nor completely different-making the previous one more likely to interfere with the current stimulus representation.

Finally, this is definitely a minor point, but I still find the reply to my comment about the prediction of stable retinal input rather speculative. Such a prediction would seem more plausible in world-centered coordinates.

---

## [Author Response]

The following is the authors’ response to the original reviews.

**Reviewer #1 (Public review):**
(1) The manuscript is quite dense, with some concepts that may prove difficult for the non-specialist. I recommend spending a few more words (and maybe some pictures) describing the difference between task-relevant and task-irrelevant planes. Nice technique, but not instantly obvious. Then we are hit with "stimulus-related", which definitely needs some words (also because it is orthogonal to neither of the above).

We agree that the original description of the planes was too terse and have expanded on this in the revised manuscript.

Line 85 - To test the influence of attention, trials were sorted according to two spatial reference planes, based on the location of the stimulus: task-related and task-unrelated (Fig. 1b). The task-related plane corresponded to participants’ binary judgement (Fig 1b, light cyan vertical dashed line) and the task-unrelated plane was orthogonal to this (Fig 1b, dark cyan horizontal dashed line). For example, if a participant was tasked with performing a left-or-right of fixation judgement, then their task-related plane was the vertical boundary between the left and right side of fixation, while their task-unrelated plane was the horizontal boundary. The former (left-right) axis is relevant to their task while the latter (top-bottom) axis is orthogonal and task irrelevant. This orthogonality can be leveraged to analyze the same data twice (once according to the task-related plane and again according to the taskunrelated plane) in order to compare performance when the relative location of an event is either task relevant or irrelevant.

Line 183 - whereas task planes were constant, the stimulus-related plane was defined by the location of the stimulus on the previous trial, and thus varied from trial to trial. That is, on each trial, the target is considered a repeat if it changes location by <|90°| relative to its location on the previous trial, and an alternate if it moves by >|90°|.

(2) While I understand that the authors want the three classical separations, I actually found it misleading. Firstly, for a perceptual scientist to call intervals in the order of seconds (rather than milliseconds), "micro" is technically coming from the raw prawn. Secondly, the divisions are not actually time, but events: micro means one-back paradigm, one event previously, rather than defined by duration. Thirdly, meso isn't really a category, just a few micros stacked up (and there's not much data on this). And macro is basically patterns, or statistical regularities, rather than being a fixed time. I think it would be better either to talk about short-term and long-term, which do not have the connotations I mentioned. Or simply talk about "serial dependence" and "statistical regularities". Or both.

We agree that the temporal scales defined in the current study are not the only way one could categorize perceptual time. We also agree that by using events to define scales, we ignore the influence of duration. In terms of the categories, we selected these for two reasons: (1) they conveniently group previous phenomena, and (2) they loosely correspond to iconic-, short- and long-term memory. We agree that one could also potentially split it up into two categories (e.g., short- and long-term), but in general, we think any form of discretization will have limitations. For example, Reviewer 1 suggests that the meso category is simply a few micros stacked together. However, there is a rich literature on phenomena associated with sequences of an intermediate length that do not appear to be entirely explained by stacking micro effects (e.g., sequence learning and sequential dependency). We also find that when controlling for micro level effects, there are clear meso level effects. Also, by the logic that meso level effects are just stacked micro effects, one could also argue the same for macro effects. We don’t think this argument is incorrect, rather we think it exemplifies the challenge of discretising temporal scales. Ultimately, the current study was aimed to test whether seemingly disparate phenomena identified in previous work could be captured by unifying principles. To this end we found that these categories were the most useful. However, we have included a “Limitations and future directions” section in the Discussion of the revised manuscript that acknowledges both the alternative scheme proposed by Reviewer 1, and the value of extending this work to consider the influence of duration (as well as events).

Line 488 - Limitations and future directions. One potential limitation of the current study is the categorization of temporal scales according to events, independent of the influence of event duration. While this simplification of time supports comparison between different phenomena associated with each scale (e.g., serial dependence, sequential dependencies, statistical learning), future work could investigate the role of duration to provide a more comprehensive understanding of the mechanisms identified in the current study.

Related to this, while the temporal scales applied here conveniently categorized known sensory phenomena, and partially correspond to iconic-, short-, and long-term memory, they are but one of multiple ways to delineate time. For example, temporal scales could alternatively be defined simply as short- and long-term (e.g., by combining micro and meso scale phenomena). However, this could obscure meaningful differences between phenomena associated with sensory persistence and short-term memory, or qualitative differences in the way that shortsequences of events are processed.

(3) More serious is the issue of precision. Again, this is partially a language problem. When people use the engineering terms "precision" and "accuracy" together, they usually use the same units, such as degrees. Accuracy refers to the distance from the real position (so average accuracy gives bias), and precision is the clustering around the average bias, usually measured as standard deviation. Yet here accuracy is percent correct: also a convention in psychology, but not when contrasting accuracy with precision, in the engineering sense. I suggest you change "accuracy" to "percent correct". On the other hand, I have no idea how precision was defined. All I could find was: "mixture modelling was used to estimate the precision and guess rate of reproduction responses, based on the concentration (k) and height of von Mises and uniform distributions, respectively". I do not know what that means.

In the case of a binary decision, is seems reasonable to use the term “accuracy” to refer to the correspondence between the target state and the response on a task. However, we agree that while our (main) task is binary, the target is not and nor is the secondary task. We thank the reviewer for bringing this to our attention, as we agree that this will be a likely cause of confusion. To avoid confusion we have specifically referred to “task accuracy” throughout the revised manuscript.

With regards to precision, our measure of precision is consistent with what Reviewer 1 describes as such, i.e., the clustering of responses. In particular, the von Mises distribution is essentially a Gaussian distribution in circular space, and the kappa parameter defines the width of the distribution, regardless of the mean, with larger values of kappa indicating narrower (more precise) distributions. We could have used standard deviation to assess precision; however, this would incorrectly combine responses on which participants failed to encode the target (e.g., because of a blink) and were simply guessing. To account for these trials, we applied mixture modelling of guess and genuine responses to isolate the precision of genuine responses, as is standard in the visual working memory literature. However, we agree that this was not sufficiently described in the original manuscript and have elaborated on this method in the revised version.

Line 598 - From the reproduction task, we sought to estimate participant’s recall precision. It is likely that on some trials participants failed to encode the target and were forced to make a response guess. To isolate the recall precision from guess responses, we used mixture modelling to estimate the precision and guess rate of reproduction responses, based on the concentration (k) and height of von Mises and uniform distributions, respectively (Bays et al., 2009). The k parameter of the von Mises distribution reflects its width, which indicates the clustering of responses around a common location.

(4) Previous studies show serial dependence can increase bias but decrease scatter (inverse precision) around the biased estimate. The current study claims to be at odds with that. But are the two measures of precision relatable? Was the real (random) position of the target subtracted from each response, leaving residuals from which the inverse precision was calculated? (If so, the authors should say so..) But if serial dependence biases responses in essentially random directions (depending on the previous position), it will increase the average scatter, decreasing the apparent precision.

Previous studies have shown that when serial dependence is attractive there is a corresponding increase in precision around small offsets from the previous item (citations). Indeed, attractive biases will lead to reduced scattering (increased precision) around a central attracter. Consistent with previous studies, and this rational, we also found an attractive bias coupled with increased precision. To clarify, for the serial dependency analysis, we calculated bias and precision by binning reproduction responses according to the offset between the current and previous target and then performing the same mixture modelling described above to estimate the mean (bias) and kappa (precision) parameters of the von Mises distribution fit to the angular errors. This was not explained in the original manuscript, so we thank Reviewer 1 for bringing this to our attention and have clarified the analysis in the revised version.

Line 604 - For the serial dependency analysis, we calculated bias and precision by binning reproduction responses according to the angular offset between the current and previous target and then performing mixture modelling to estimate the mean (bias) and k (precision) parameters of the von Mises distribution.

(5) I suspect they are not actually measuring precision, but location accuracy. So the authors could use "percent correct" and "localization accuracy". Or be very clear what they are actually doing.

As explained in our response to Reviewer 1’s previous comment, we are indeed measuring precision.

**Reviewer #2 (Public review):**
(1) The abstract should more explicitly mention that conclusions about feedforward mechanisms were derived from a reanalysis of an existing EEG dataset. As it is, it seems to present behavioral data only.

It is not clear what relevance the fact that the data has been analyzed previously has to the results of the current study. However, we do think that it is important to be clear that the EEG recordings were collected separately from the behavioural and eyetracking data, so we have clarified this in the revised abstract.

Line 7 - By integrating behavioural and pupillometry recordings with electroencephalographical recordings from a previous study, we identify two distinct mechanisms that operate across all scales.

(2) The EEG task seems quite different from the others, with location and color changes, if I understand correctly, on streaks of consecutive stimuli shown every 100 ms, with the task involving counting the number of target events. There might be different mechanisms and functions involved, compared to the behavioral experiments reported.

As stated above, we agree that it is important that readers are aware that the EEG recordings were collected separately to the behavioural and eyetracking data. We were forthright about this in the original manuscript and how now clarified this in the revised abstract. We agree that collecting both sets of data in the same experiment would be a useful validation of the current results and have acknowledged this in a new Limitations and future directions section of the Discussion of the revised manuscript.

Line 501 - Another limitation of the current study is that the EEG recordings were collected in the separate experiment to the behavioural and pupillometry data. The stimuli and task were similar between experiments, but not identical. For example, the EEG experiment employed coloured arc stimuli presented at a constant rate of ~3.3 Hz and participants were tasked with counting the number of stimuli presented at a target location. By contrast, in the behavioural experiment, participants viewed white blobs presented at an average rate of ~2.8 Hz and performed a binary spatial task coupled with an infrequent reproduction task. An advantage of this was that the sensory responses to stimuli in the EEG recordings were not conflated with motor responses; however, future work combining these measures in the same experiment would serve as a validation for the current results.

(3) How is the arbitrary choice of restricting EEG decoding to a small subset of parieto-occipital electrodes justified? Blinks and other artifacts could have been corrected with proper algorithms (e.g., ICA) (Zhang & Luck, 2025) or even left in, as decoders are not necessarily affected by noise. Moreover, trials with blinks occurring at the stimulus time should be better removed, and the arbitrary selection of a subset of electrodes, while reducing the information in input to the decoder, does not account for trials in which a stimulus was missed (e.g., due to blinks).

Electrode selection was based on several factors: (1) reduction of eye movement/blink artifacts (as noted in the original manuscript), (2) consistency with the previous EEG study (Rideaux, 2024) and other similar decoding studies (Buhmann et al., 2024; Harrison et al., 2023; Rideaux et al., 2023), (3) improved signal-to-noise by including only sensors that carry the most position information (as shown in Supplementary Figure 1a and the previous EEG study). We agree that this was insufficiently explained in the original manuscript and have clarified our sensor selection in the revised version.

Line 631 - We only included the parietal, parietal-occipital, and occipital sensors in the analyses to (i) reduce the influence of signals produced by eye movements, blinks, and non-sensory cortices, (ii) for consistency with similar previous decoding studies (Buhmann et al., 2024; Rideaux, 2024; Rideaux et al., 2025), and (iii) to improve decoding accuracy by restricting sensors to those that carried spatial position information (Supplementary Fig. 1a).

(4) The artifact that appears in many of the decoding results is puzzling, and I'm not fully convinced by the speculative explanation involving slow fluctuations. I wonder if a different high-pass filter (e.g., 1 Hz) might have helped. In general, the nature of this artifact requires better clarification and disambiguation.

We agree that the nature of this artifact requires more clarification and disambiguation. Due to relatively slow changes in the neural signal, which are not stimulus-related, there is a degree of temporal autocorrelation in the recordings. This can be filtered out, for example, by using a stricter high-pass filter; however, we tried a range of filters and found that a cut-off of at least 0.7 Hz is required to remove the artifact, and even a filter of 0.2 Hz introduces other (stimulus-related) artifacts, such as above-chance decoding prior to stimulus onset. These stimulus-related artifacts are due to the temporal smearing of data, introduced by the filtering, and have a more pronounced and complex influence on the results and are more difficult to remove through other means, such as the baseline correction applied in the original manuscript.

The temporal autocorrelation is detected by the decoder during training and biases it to classify/decode targets that are presented nearby in time as similar. That is, it learns the neural pattern for a particular stimulus location based on the activity produced by the stimulus and the temporal autocorrelation (determined by slow stimulus unrelated fluctuations). The latter only accounts for a relatively smaller proportion of the variance in the neural recordings under normal circumstances and would typically go undetected when simply plotting decoding accuracy as a function of position. However, it becomes weakly visible when decoding accuracy is plotted as a function of distance from the previous target, as now the bias (towards temporally adjacent targets) aligns with the abscissa. Further, it becomes highly visible when the stimulus labels are shuffled, as now the decoder can only learn from the variance associated with the temporal autocorrelation (and not from the activity produced by the stimulus).

In the linear discriminant analysis, this led to temporally proximal items being more likely to be classified as on the same side. This is why there is above-chance performance for repeat trials (Supplementary Figure 2b), and below-chance performance for alternate trials, even when the labels are shuffled – the temporal autocorrelation produces a general bias towards classifying temporally proximate stimuli as on the same side, which selectively improves the classification accuracy of repeat trials. Fortunately, the bias is relatively constant as a function of time within the epoch and is straightforward to estimate by shuffling the labels, which means that it can be removed through a baseline correction. However, to further demonstrate that the autocorrelation confound cannot account for the differences observed between repeat and alternate trials in the micro classification analysis, we now additionally show the results from a more strictly filtered version of the data (0.7 Hz). These results show a similar pattern as the original, with the additional stimulusrelated artifacts introduced by the strict filter, e.g., above chance decoding prior to stimulus onset.

In the inverted encoding analysis, the same temporal autocorrelation manifests as temporally proximal trials being decoded as more similar locations. This is why there is increased decoding accuracy for targets with small angular offsets from the previous target, even when the labels are shuffled (Supplementary Figure 3c), because it is on these trials that the bias happens to align with the correct position. This leads to an attractive bias towards the previous item, which is most prominent when the labels are shuffled.

To demonstrate the phenomenon, we simulated neural recordings from a population of tuning curves and performed the inverted encoding analysis on a clean version of the data and a version in which we introduced temporal autocorrelation. We then repeated this after shuffling the labels. The simulation produced very similar results to those we observed in the empirical data, with a single exception: while precision in the simulated shuffled data was unaffected by autocorrelation, precision in the unshuffled data was clearly affected by this manipulation. This may explain why we did not find a correlation between the shuffled and unshuffled precision in the original manuscript.

These results echo those from the classification analysis, albeit in a more continuous space. However, whereas in the classification analysis it was straightforward to perform a baseline correction to remove the influence of general temporal dependency, the more complex nature of the accuracy, precision, and bias parameters over the range of time and delta location makes this approach less appropriate. For example, the bias in the shuffled condition ranged from -180 to 180 degrees, which when subtracted from the bias in the unshuffled condition would produce an equally spurious outcome, i.e., the equal opposite of this extreme bias. Instead for the inverted encoding analysis, we used the data high-pass filtered at 0.7 Hz. As with the classification analysis, this removed the influence of general temporal dependencies, as indicated by the results of the shuffled data analysis (Supplementary Figure 3f), but it also temporally smeared the stimulus-related signal, resulting in above chance decoding accuracy prior to stimulus onset (Supplementary Figure 3d). However, given thar we were primarily interested in the pattern of accuracy, precision, and bias as a function of delta location, and less concerned with the precise temporal dynamics of these changes, which appeared relatively stable in the filtered data. Thus, this was the more suitable approach to removing the general temporal dependencies in the inverted encoding analysis and the one that is presented in Figure 3.

We have updated the revised manuscript in light of these changes, including a fuller description of the artifact and the results from the abovementioned control analyses.

Figure 3 updated.

Figure 3 caption - (e) Decoding accuracy for stimulus location, from reanalysis of previously published EEG data (17). Inset shows the EEG sensors included in the analysis (blue dots), and black rectangles indicate the timing of stimulus presentations (solid: target stimulus, dashed: previous and subsequent stimuli). (f) Decoding accuracy for location, as a function of time and D location. Bright colours indicate higher decoding accuracy; absolute accuracy values can be inferred from (e). (g-i) Average location decoding (g) accuracy, (h) precision, and (h) bias from 50 – 500 ms following stimulus onset. Horizontal bar in (e) indicates cluster corrected periods of significance; note, all time points were significantly above chance due to temporal smear introduced by strict high-pass filtering (see Supplementary Figure 3 for full details). Note, the temporal abscissa is aligned across (e & f). Shaded regions indicate ± SEM.

Line 218 - To further investigate the influence of serial dependence, we applied inverted encoding modelling to the EEG recordings to decode the angular location of stimuli. We found that decoding accuracy of stimulus location sharply increased from ~60 ms following stimulus onset (Fig. 3e). Note, to reduce the influence of general temporal dependencies, we applied a 0.7 Hz high-pass filter to the data, which temporally smeared the stimulus-related information, resulting in above chance decoding accuracy prior to stimulus presentation (for full details, see Supplementary Figure 3). To understand how serial dependence influences the representation of these features, we inspected decoding accuracy for location as a function of both time and D location (Fig. 3f). We found that decoding accuracy varied depending not only as a function of time, but also as a function of D location. To characterise this relationship, we calculated the average decoding accuracy from 50 ms until the end of the epoch (500 ms), as a function of D location (Fig. 3g). This revealed higher accuracy for targets with larger D location. We found a similar pattern of results for decoding precision (Fig. 3h). These results are consistent with the micro temporal context (behavioural) results, showing that targets that alternated were recalled more precisely. Lastly, we calculated the decoding bias as a function of D location and found a clear repulsive bias away from the previous item (Fig. 3i). While this result is inconsistent with the attractive behavioural bias, it is consistent with recent studies of serial dependence suggesting an initial pattern of repulsion followed by an attractive bias during the response period (20–22).

Line 726 - As shown in Supplementary Figure 3, we found the same general temporal dependencies in the decoding accuracy computed using inverted encoding that were found using linear discriminant classification. However, as a baseline correction would not have been appropriate or effective for the parameters decoded with this approach, we instead used a high-pass filter of 0.7 Hz to remove the confound, while being cautious about interpreting the timing of effects produced by this analysis due to the temporal smear introduced by the filter.

Supplementary Figure 2 updated.

Supplementary Figure 2 caption - Removal of general micro temporal dependencies in EEG responses. We found that there were differences in classification accuracy for repeat and alternate stimuli in the EEG data, even when stimulus labels were shuffled. This is likely due to temporal autocorrelation within the EEG data due to low frequency signal changes that are unrelated to the decoded stimulus dimension. This signal trains the decoder to classify temporally proximal stimuli as the same class, leading to a bias towards repeat classification. For example, in general, the EEG signal during trial one is likely to be more similar to that during trial two than during trial ten, because of low frequency trends in the recordings. If the decoder has been trained to classify the signal associated with trial one as a leftward stimulus, then it will be more likely to classify trial two as a leftward stimulus too. These autocorrelations are unrelated to stimulus features; thus, to isolate the influence of stimulus-specific temporal context, we subtracted the classification accuracy produced by shuffling the stimulus labels from the unshuffled accuracy (as presented in Figure 2e, f). We confirmed that using a stricter high-pass filter (0.7 Hz) removes this artifact, as indicated by the equal decoding accuracy between the two shuffled conditions. However, the stricter high-pass filter temporally smears the stimulus-related signal, which introduces other (stimulus-related) artifacts, e.g., above-chance decoding accuracy prior to stimulus presentation, that are larger and more complex, i.e., changing over time. Thus, we opted to use the original high pass filter (0.1 Hz) and apply a baseline correction. (a) The uncorrected classification accuracy along task related and unrelated planes. Note that these results are the same as the corrected version shown in Figure 2e, because the confound is only apparent when accuracy is grouped according to temporal context.

(b) Same as (a), but split into repeat and alternate stimuli, along (left) task-related and (right) unrelated planes. Classification accuracy when labels are shuffled is also shown. Inset in (a) shows the EEG sensors included in the analysis (blue dots). (c, d) Same as (a, b), but on data filtered using a 0.7 Hz high-pass filter. Black rectangles indicate the timing of stimulus presentations (solid: target stimulus, dashed: previous and subsequent stimuli). Shaded regions indicate ± SEM.

Supplementary Figure 3 updated.

Supplementary Figure 3 caption - Removal of general temporal dependencies in EEG responses for inverted encoding analyses. As described in Methods - Neural Decoding, we used inverted encoding modelling of EEG recordings to estimate the decoding accuracy, precision, and bias of stimulus location. Just as in the linear discriminant classification analysis, we also found the influence of general temporal dependencies in the results produced by the inverted encoding analysis. In particular, there was increased decoding accuracy for targets with low D location. This was weakly evident in the period prior to stimulus presentation, but clearly visible when the labels were shuffled. These results are mirror those from the classification analysis, albeit in a more continuous space. However, whereas in the classification analysis it was straightforward to perform a baseline correction to remove the influence of general temporal dependency, the more complex nature of the accuracy, precision, and bias parameters over the range of time and D location makes this approach less appropriate. For example, the bias in the shuffled condition ranged from -180° to 180°, which when subtracted from the bias in the unshuffled condition would produce an equally spurious outcome, i.e., the equal opposite of this extreme bias. Instead for the inverted encoding analysis, we used the data high-pass filtered at 0.7 Hz. As with the classification analysis, this significantly reduced the influence of general temporal dependencies, as indicated by the results of the shuffled data analysis, but it also temporally smeared the stimulus-related signal, resulting in above chance decoding accuracy prior to stimulus onset. However, we were primarily interested in the pattern of accuracy, precision, and bias as a function of D location, and less concerned with the precise temporal dynamics of these changes. Thus, this was the more suitable approach to removing the general temporal dependencies in the inverted encoding analysis and the one that is presented in Figure 3. (a) Decoding accuracy as a function of time for the EEG data filtered using a 0.1 Hz high-pass filter. Inset shows the EEG sensors included in the analysis (blue dots), and black rectangles indicate the timing of stimulus presentations (solid: target stimulus, dashed: previous and subsequent stimuli). (b, c) The same as (a), but as a function of time and D location for (b) the original data and (c) data with shuffled labels. (d-f) Same as (a-c), but for data filtered using a 0.7 Hz high-pass filter. Shaded regions in (a, d) indicate ± SEM. Horizontal bars in (a, d) indicate cluster corrected periods of significance; note, all time points in (d) were significantly above chance. Note, the temporal abscissa is vertically aligned across plots (a-c & d-f).

In the process of performing these additional analyses and simulations, we became aware that the sign of the decoding bias in the inverted encoding analyses had been interpreted in the wrong direction. That is, where we previously reported an initial attractive bias followed by a repulsive bias relative to the previous target, we have in fact found the opposite, an initial repulsive bias followed by an attractive bias relative to the previous target. Based on the new control analyses and simulations, we think that the latter attractive bias was due to general temporal dependencies. That is, in the filtered data, we only observe a repulsive bias. While the bias associated with serial dependence was not a primary feature of the study, this (somewhat embarrassing) discovery has led to reinterpretation of some results relating to serial dependence. However, it is encouraging to see that our results now align with those of recent studies (Fischer et al., 2024; Luo et al., 2025; Sheehan et al. 2024).

Line 385 - Our corresponding EEG analyses revealed better decoding accuracy and precision for stimuli preceded by those that were different and a bias away from the previous stimulus. These results are consistent with finding that alternating stimuli are recalled more precisely. Further, while the repulsive pattern of biases is inconsistent with the observed behavioural attractive biases, it is consistent with recent work on serial dependence indicating an initial period of repulsion, followed by an attractive bias during the response period (20–22). These findings indicate that serial dependence and first-order sequential dependencies can be explained by the same underlying principle.

(5) Given the relatively early decoding results and surprisingly early differences in decoding peaks, it would be useful to visualize ERPs across conditions to better understand the latencies and ERP components involved in the task.

A rapid presentation design was used in the EEG experiment, and while this is well suited to decoding analyses, unfortunately we cannot resolve ERPs because the univariate signal is dominated by an oscillation at the stimulus presentation frequency (~3 Hz). We agree that this could be useful to examine in future work.

(6) It is unclear why the precision derived from IEM results is considered reliable while the accuracy is dismissed due to the artifact, given that both seem to be computed from the same set of decoding error angles (equations 8-9).

This point has been addressed in our response to point (4).

(7) What is the rationale for selecting five past events as the meso-scale? Prior history effects have been shown to extend much further back in time (Fritsche et al., 2020).

We used five previous items in the meso analyses to be consistent with previous research on sequential dependencies (Bertelson, 1961; Gao et al., 2009; Jentzsch & Sommer, 2002; Kirby, 1976; Remington, 1969). However, we agree that these effects likely extend further and have acknowledged this in the revied version of the manuscript.

Line 240 - Higher-order sequential dependences are an example of how stimuli (at least) as far back as five events in the past can shape the speed and task accuracy of responses to the current stimulus (9, 10); however, note that these effects have been observed for more than five events (20).

(8) The decoding bias results, particularly the sequence of attraction and repulsion, appear to run counter to the temporal dynamics reported in recent studies (Fischer et al., 2024; Luo et al., 2025; Sheehan & Serences, 2022).

This point has been addressed in our response to point (4).

(9) The repulsive component in the decoding results (e.g., Figure 3h) seems implausibly large, with orientation differences exceeding what is typically observed in behavior.

As noted in our response to point (4), this bias was likely due to the general temporal dependency confound and has been removed in the revised version of the manuscript.

(10) The pattern of accuracy, response times, and precision reported in Figure 3 (also line 188) resembles results reported in earlier work (Stewart, 2007) and in recent studies suggesting that integration may lead to interference at intermediate stimulus differences rather than improvement for similar stimuli (Ozkirli et al., 2025).

Thank you for bringing this to our attention, we have acknowledged this in the revised manuscript.

Line 197 - Consistent with our previous binary analysis, and with previous work (19), we also found that responses were faster and more accurate when D location was small (Fig. 3b, c).

(11) Some figures show larger group-level variability in specific conditions but not others (e.g., Figures 2b-c and 5b-c). I suggest reporting effect sizes for all statistical tests to provide a clearer sense of the strength of the observed effects.

Yes, as noted in the original manuscript, we find significant differences between the variance task-related and -unrelated conditions. We think this is due to opposing forces in the task-related condition:

“The increased variability of response time differences across the taskrelated plane likely reflects individual differences in attention and prioritization of responding either quickly or accurately. On each trial, the correct response (e.g., left or right) was equally probable. So, to perform the task accurately, participants were motivated to respond without bias, i.e., without being influenced by the previous stimulus. We would expect this to reduce the difference in response time for repeat and alternate stimuli across the taskrelated plane, but not the task-unrelated plane. However, attention may amplify the bias towards making faster responses for repeat stimuli, by increasing awareness of the identity of stimuli as either repeats or alternations (17). These two opposing forces vary with task engagement and strategy and thus would be expected produce increased variability across the task-related plane.” We agree that providing effect sizes may provided a clearer sense of the observed effects and have done so in the revised version of the manuscript.

Line 739 - For Wilcoxon signed rank tests, the rank-biserial correlation (r) was calculated as an estimate of effect size, where 0.1, 0.3, and 0.5 indicate small, medium, and large effects, respectively (54). For Friedman’s ANONA tests, Kendal’s W was calculated as an estimate of effect size, where 0.1, 0.3, and 0.5 indicate small, medium, and large effects, respectively (55).

(12) The statement that "serial dependence is associated with sensory stimuli being perceived as more similar" appears inconsistent with much of the literature suggesting that these effects occur at post-perceptual stages (Barbosa et al., 2020; Bliss et al., 2017; Ceylan et al., 2021; Fischer et al., 2024; Fritsche et al., 2017; Sheehan & Serences, 2022).

In light of the revised analyses, this statement has been removed from the manuscript.

(13) If I understand correctly, the reproduction bias (i.e., serial dependence) is estimated on a small subset of the data (10%). Were the data analyzed by pooling across subjects?

The dual reproduction task only occurred on 10% of trials. There were approximately 2000 trials, so ~200 reproduction responses. For the micro and macro analyses, this was sufficient to estimate precision within each of the experimental conditions (repeat/alternate, expected/unexpected). However, it is likely that we were not able to reproduce the effect of precision at the meso level across both experiments because we lacked sufficient responses to reliably estimate precision when split across the eight sequence conditions. Despite this, the data was always analysed within subjects.

(14) I'm also not convinced that biases observed in forced-choice and reproduction tasks should be interpreted as arising from the same process or mechanism. Some of the effects described here could instead be consistent with classic priming.

We agree that the results associated with the forced-choice task (response time task accuracy) were likely due to motor priming, but that a separate (predictive) mechanism may explain the (precision) results associated with the reproduction task. These are two mechanisms we think are operating across the three temporal scales investigated in the current study.

**Reviewing Editor Comments:**
(1) Clarify task design and measurement: The dense presentation makes it difficult to understand key design elements and their implications. Please provide clearer descriptions of all task elements, and how they relate to each other (EEG vs. behaviour, stimulus plane vs. TR and TU plane, reproduction vs. discrimination and role of priming), and clearly explain how key measures were computed for each of these (e.g., precision, accuracy, reproduction bias).

In the revised manuscript, we have expanded on descriptions of the source and nature of the data (behavioural and EEG), the different planes analyzed in the behavioural task, and how key metrics (e.g., precision) were computed.

(2) Offer more insight into underlying data, including original ERP waveforms to aid interpretation of decoding results and the timing of effects. In particular, unpack the decoding temporal confound further.

In the revised manuscript, we have considerably offered more insight into the decoding results, in particular, the nature of the temporal confound. We were unable to assess ERPs due to the rapid presentation design employed in the EEG experiment.

(3) Justify arbitrary choices such as electrode selection for EEG decoding (e.g., limiting to parieto-occipital sensors), number of trials in meso scale, and the time terminology itself.

In the revised manuscript, we have clarified the reasons for electrode selection.

(3) Discuss deviations from literature: Several findings appear to contradict or diverge from previous literature (e.g., effects of serial dependence). These discrepancies could be discussed in more depth.

Upon re-analysis of the serial dependence bias and removal of the temporal confound, the results of the revised manuscript now align with those from previous literature, which has been acknowledged.

**Reviewer #1 (Recommendations for the authors):**
(1) would like to use my reviewer's prerogative to mention a couple of relevant publications.Galluzzi et al (Journal of Vision, 2022) "Visual priming and serial dependence are mediated by separate mechanisms" suggests exactly that, which is relevant to this study.Xie et al. (Communications Psychology, 2025) "Recent, but not long-term, priors induce behavioral oscillations in peri-saccadic vision" also seems relevant to the issue of different mechanisms.

Thank you for bringing these studies to our attention. We agree that they are both relevant have referenced both appropriately in the revised version of the manuscript.

**Reviewer #2 (Recommendations for the authors):**
(1) I find the discussion on attention and awareness (from line 127 onward) somewhat vague and requiring clarification.

We agree that this statement was vague and referred to “awareness” without operationation. We have revised this statement to improve clarity.

Line 135 - However, task-relatedness may amplify the bias towards making faster responses for repeat stimuli, by increasing attention to the identity of stimuli as either repeats or alternations (17).

(2) Line 140: It's hard to argue that there are expectations that the image of an object on the retina is likely to stay the same, since retinal input is always changing.

We agree that retinal input is often changing, e.g., due to saccades, self-motion, and world motion. However, for a prediction to be useful, e.g., to reduce metabolic expenditure or speed up responses, it must be somewhat precise, so a prediction that retinal input will change is not necessarily useful, unless it can specify what it will change to. Given retinal input of x at time t, the range of possible values of x at time t+1 (predicting change) is infinite. By contrast, if we predict that x=x at time t+1 (no change), then we can make a precise prediction. There is, of course, other information that could be used to reduce the parameter space of predicted change from x at time t, e.g., the value of x at time t-1, and we think this drives predictions too. However, across the infinite distribution of changes from x, zero change will occur more frequently than any other value, so we think it’s reasonable to assert that the brain may be sensitive to this pattern.

(3) Line 564: The gambler's fallacy usually involves sequences longer than just one event.

Yes, we agree that this phenomenon is associated with longer sequences. This section of the manuscript was in regards to previous findings that were not directly relevant to the current study and has been removed in the revised version.

(4) In the shared PDF, the light and dark cyan colors used do not appear clearly distinguishable.

I expect this is due to poor document processing or low-quality image embeddings. I will check that they are distinguishable in the final version.

References:

Barbosa, J., Stein, H., Martinez, R. L., Galan-Gadea, A., Li, S., Dalmau, J., Adam, K. C. S., Valls-Solé, J., Constantinidis, C., & Compte, A. (2020). Interplay between persistent activity and activity-silent dynamics in the prefrontal cortex underlies serial biases in working memory. Nature Neuroscience, 23(8), Articolo 8. https://doi.org/10.1038/s41593-020-0644-4

Bliss, D. P., Sun, J. J., & D'Esposito, M. (2017). Serial dependence is absent at the time of perception but increases in visual working memory. Scientific reports, 7(1), 14739.

Ceylan, G., Herzog, M. H., & Pascucci, D. (2021). Serial dependence does not originate from low-level visual processing. Cognition, 212, 104709. https://doi.org/10.1016/j.cognition.2021.104709

Fischer, C., Kaiser, J., & Bledowski, C. (2024). A direct neural signature of serial dependence in working memory. eLife, 13. https://doi.org/10.7554/eLife.99478.1

Fritsche, M., Mostert, P., & de Lange, F. P. (2017). Opposite effects of recent history on perception and decision. Current Biology, 27(4), 590-595.

Fritsche, M., Spaak, E., & de Lange, F. P. (2020). A Bayesian and efficient observer model explains concurrent attractive and repulsive history biases in visual perception. eLife, 9, e55389. https://doi.org/10.7554/eLife.55389

Gekas, N., McDermott, K. C., & Mamassian, P. (2019). Disambiguating serial effects of multiple timescales. Journal of vision, 19(6), 24-24.

Luo, M., Zhang, H., Fang, F., & Luo, H. (2025). Reactivation of previous decisions repulsively biases sensory encoding but attractively biases decision-making. PLOS Biology, 23(4), e3003150. https://doi.org/10.1371/journal.pbio.3003150

Ozkirli, A., Pascucci, D., & Herzog, M. H. (2025). Failure to replicate a superiority effect in crowding. Nature Communications, 16(1), 1637. https://doi.org/10.1038/s41467025-56762-5

Sheehan, T. C., & Serences, J. T. (2022). Attractive serial dependence overcomes repulsive neuronal adaptation. PLoS biology, 20(9), e3001711.

Stewart, N. (2007). Absolute identification is relative: A reply to Brown, Marley, and

Lacouture (2007). Psychological Review, 114, 533-538. https://doi.org/10.1037/0033-295X.114.2.533

Treisman, M., & Williams, T. C. (1984). A theory of criterion setting with an application to sequential dependencies. Psychological review, 91(1), 68.

Zhang, G., & Luck, S. J. (2025). Assessing the impact of artifact correction and artifact rejection on the performance of SVM- and LDA-based decoding of EEG signals. NeuroImage, 316, 121304. https://doi.org/10.1016/j.neuroimage.2025.121304